

# On the implications of aerosol liquid water and phase separation for organic aerosol mass

Havala O. T. Pye[1], Benjamin N. Murphy[1], Lu Xu[2], Ng L. Ng[2,3], Annmarie G. Carlton[4,5], Hongyu Guo[3], Rodney Weber[3], Petros Vasilakos[2], K. Wyat Appel[1], Sri Hapsari Budisulistiorini[6], Jason D. Surratt[6], Athanasios Nenes[2,3,7,8], Weiwei Hu[9,10], Jose L. Jimenez[9,10], Gabriel Isaacman-VanWertz[11], Pawel K. Misztal[11], and Allen H. Goldstein[11,12]

[1]National Exposure Research Laboratory, US Environmental Protection Agency, Research Triangle Park, NC, USA
[2]School of Chemical and Biomolecular Engineering, Georgia Institute of Technology, Atlanta, GA, USA
[3]School of Earth and Atmospheric Sciences, Georgia Institute of Technology, Atlanta, GA, USA
[4]Department of Environmental Sciences, Rutgers University, New Brunswick, NJ, USA
[5]Now at: Department of Chemistry, University of California, Irvine, CA, USA
[6]Gillings School of Global Public Health, The University of North Carolina at Chapel Hill, Chapel Hill, NC, USA
[7]Institute of Environmental Research and Sustainable Development, National Observatory of Athens, Palea Penteli, GR-15236, Greece
[8]Institute for Chemical Engineering Sciences, Foundation for Research and Technology Hellas, Patras, Greece
[9]Cooperative Institute for Research in Environmental Sciences, University of Colorado, Boulder, CO, USA
[10]Department of Chemistry and Biochemistry, University of Colorado, Boulder, CO, USA
[11]Department of Environmental Science, Policy, and Management, University of California, Berkeley, CA USA
[12]Department of Civil and Environmental Engineering, University of California, Berkeley, CA USA

*Correspondence to:* Havala O. T. Pye (pye.havala@epa.gov)

**Abstract.** Organic compounds and liquid water are major aerosol constituents in the southeast United States (SE US). Water associated with inorganic constituents (inorganic water) can contribute to the partitioning medium for organic aerosol when relative humidities or organic matter to organic carbon (OM/OC) ratios are high such that separation relative humidities (SRH) are below the ambient relative humidity (RH). As OM/OC ratios in the SE US are often between 1.8 and 2.2, organic aerosol

experiences both mixing with inorganic water and separation from it. Regional chemical transport model simulations including inorganic water (but excluding water uptake by organic compounds) in the partitioning medium for secondary organic aerosol (SOA) when RH>SRH led to increased SOA concentrations, particularly at night. Water uptake to the organic phase resulted in even greater SOA concentrations as a result of a positive feedback in which water uptake increased SOA, which further increased aerosol water and organic aerosol. Aerosol properties, such as the OM/OC and hygroscopicity parameter ($\kappa_{org}$),

were captured well by the model compared with measurements during the Southern Oxidant and Aerosol Study (SOAS) 2013. Organic nitrates from monoterpene oxidation were predicted to be the least water-soluble semivolatile species in the model, but most biogenically-derived semivolatile species in the CMAQ model were highly water soluble, and expected to contribute to water soluble organic carbon (WSOC). Organic aerosol and SOA precursors were abundant at night; but, additional improvements in daytime organic aerosol are needed to close the model-measurement gap. By taking into account deviations

from ideality, including both inorganic (when RH>SRH) and organic water in the organic partitioning medium reduced the



mean bias in SOA for routine monitoring networks and improved model performance compared to observations from SOAS. Property updates from this work will be released in CMAQ v5.2.

## 1 Introduction

Water is a ubiquitous component of atmospheric aerosol (Nguyen et al., 2016) that can interact with organic compounds in a number of ways to influence particulate matter (PM) mass and size, human health, and Earth's radiative balance. While constituents such as sulfate and nitrate often drive aerosol water concentrations, inorganic and organic compounds are internally mixed under humid conditions (You et al., 2013), and hydrophilic organic compounds promote the uptake of water (Saxena et al., 1995). Uptake of water onto the organic phase increases particle size, making particles more effective at interacting with radiation (Chung and Seinfeld, 2002), obscuring visibility (Lowenthal and Kumar, 2016), and forming clouds (Novakov and Penner, 1993). Water can serve as a medium for partitioning of soluble (Carlton and Turpin, 2013; Pun et al., 2002) and semivolatile (Chang and Pankow, 2010; Pankow and Chang, 2008; Seinfeld et al., 2001) gases, thus contributing to particulate matter concentrations. Once in the particle-phase, organic compounds can participate in water-mediated reactions such as hydrolysis, driving isoprene epoxydiol uptake to the particle (Surratt et al., 2010; Pye et al., 2013) and loss of gas-phase organic nitrates (Fisher et al., 2016).

Organic aerosol-water interactions have been examined in a number of laboratory and field studies, and results are mixed. The lack of a consistent relationship in laboratory work may be partially due to experimental conditions such as high mass loadings that minimize the effect of water on semivolatile systems (Pankow and Chang, 2008). Laboratory studies have observed no significant change in yield with increasing relative humidity (RH) (Edney et al., 2000; Boyd et al., 2015), enhanced yields under dry conditions (Zhang et al., 2011), and higher yields with increasing aerosol water (Wong et al., 2015) depending on the precursor, oxidant, and seed. Trends in ambient aerosol organic carbon are consistent with the trend in decreasing aerosol water in the southeast U.S. (Nguyen et al., 2015b), and Hennigan et al. (2008) observed episodic correlations of water-soluble organic carbon and water vapor. However, Hennigan et al. (2008) found no well-defined relationship over the entire summer in Atlanta, GA, and organic aerosol was not correlated with liquid water content in Pittsburgh, PA (Griffin et al., 2003). Saxena et al. (1995) found that the presence of organic compounds suppressed aerosol water in urban locations. In the atmosphere, the relative roles of different secondary organic aerosol (SOA) species change as a function of time and space and each species may have a different sensitivity to aerosol water.

The interaction of primary organic aerosol (POA), SOA from low-volatility and semivolatile ($C_i^* < 3000$ µgm$^{-3}$) compounds, SOA from aqueous pathways, and the inorganic/water-rich phase is important for the concentration of organic aerosol (OA) as partitioning is a function of the availability of an absorptive medium. Current chemical transport models, including the Community Multiscale Air Qualilty (CMAQ) model (Carlton et al., 2010), consider SOA to be exclusively or dominantly formed via condensation of organic compounds in the absence of water. Individual model studies have examined hydrophobic and hydrophilic SOA through semi-mechanistic algorithms and surrogate structure information. Pun et al. (2002) used a decoupled approach in which organic species partitioned only to their dominant phase (aqueous vs organic). Griffin et al. (2003)



allowed compounds to partition to both phases in varying amounts based on their properties. Jathar et al. (2016) examined the implications of water uptake to the organic phase and the effects on OA concentrations. Pun (2008) allowed organic compounds to interact with water and separate into two phases if thermodynamically favorable. None of these approaches considered mixing of the inorganic and organic phases and often required computationally intensive calculations of activity coefficients. These

models accounting for aerosol water-organic interactions are not in widespread use and have not been evaluated with recently available observations of aerosol water.

Figure 1 shows the contribution of POA and water-soluble OA (determined from water soluble organic carbon, WSOC (Sullivan et al., 2004)) to total OA as observed during the Southern Oxidant and Aerosol Study (SOAS) for June 2013 in Centreville, AL. Ambient measurements of WSOC are highly correlated with oxygenated organic aerosol (OOA) (Kondo et al.,

2007), and water-soluble OA accounted for 90% of total OA on average in the southeast US during summer 2013 (Washenfelder et al., 2015). WSOC has also been proposed to represent SOA from aqueous pathways with evidence for reversible (El-Sayed et al., 2015) and irreversible (El-Sayed et al., 2016) formation. CMAQ tends to over-predict the concentration of POA by almost a factor of 2 during SOAS (Pye et al., 2015). CMAQ predicts a relatively minor role for aqueous OA with the dominant source of OA in CMAQ being dry processes (other SOA in Figure 1).

This work aims to understand if interactions of aerosol water with semivolatile compounds can resolve model-measurement discrepancies and to what degree OA predicted by models should be classified as water soluble. Semi-empirical SOA in the CMAQ model was connected to a consistent set of properties useful for predicting atmospherically relevant behavior such as interaction with aerosol water. In cases where a specific molecular species was not already used as a surrogate, aerosol properties were linked to volatility and parent hydrocarbon (Section 2.2). These quantities allowed molecular weights, organic

matter to organic carbon (OM/OC) ratios, Henry's law coefficients, deposition properties, hygroscopicity ($\kappa_i$), phase separation (Section 2.3), water uptake (Section 2.4), and deviations from ideality (Section 2.6) to be predicted semi-empirically. The implications of the updates for OA and particle-phase liquid water content (LWC) are examined in the context of routine monitoring networks and SOAS observations.

## 2   Method

### 2.1   CMAQ organic aerosol

CMAQ v5.1 contains several types of SOA with different sensitivities to aerosol water: traditional semivolatile SOA from Odum 2-product representations, nonvolatile SOA produced by volatile organic compound (VOC) reaction, heterogeneously produced SOA parameterized by an uptake coefficient, semivolatile organic nitrate SOA and its hydrolysis product, and other contributions from cloud processing and accretion/oligomerization reactions (Figure 2, Table 1). The traditional SOA systems

in CMAQ include SOA from isoprene, monoterpenes, sesquiterpenes, benzene, toluene, xylene, alkanes, and polycyclic aromatics hydrocarbons (PAHs) (Carlton et al., 2010; Pye and Pouliot, 2012). The semivolatile SOA from these precursors is allowed to oligomerize to a nonvolatile form on a 29-hour timescale (Carlton et al., 2010). Currently, low-NO$_x$ oxidation of aromatics leads to nonvolatile SOA in the traditional systems. Glyoxal, methylglyoxal, and epoxides undergo heterogeneous





uptake to form SOA (Pye et al., 2013, 2015). Glyoxal SOA forms using a fixed uptake coefficient of 0.0029 (Liggio et al., 2005). Following the approach of Marais et al. (2016), methylglyoxal's uptake coefficient was scaled to the glyoxal uptake coefficient by the relative Henry's law coefficient (resulting in an uptake coefficient of $2.6 \times 10^{-4}$) in this work. Isoprene epoxydiol (IEPOX) SOA is parameterized with an uptake coefficient calculated as a function of aerosol-phase constituents, including sulfate and water assuming an acid-catalyzed mechanism (Pye et al., 2013). In this work, the IEPOX organosulfate formation rate constant was updated to $8.83 \times 10^{-3} \ \mathrm{M^{-2}s^{-1}}$ using the ratio of 2-methyltetrol to organosulfate formation rate constants from Piletic et al. (2013) and a 2-methyltetrol rate constant of $9 \times 10^{-4} \ \mathrm{M^{-2}s^{-1}}$ (Eddingsaas et al., 2010). This organosulfate rate constant is more aggressive (overall and relative) than predicted by Riedel et al. (2016). Overestimates of the organosulfate in the model may compensate for missing IEPOX-derived SOA species such as $C_5$-alkene triols (Surratt et al., 2010) or additional oligomers (Lopez-Hilfiker et al., 2016). In addition, the Henry's law coefficient was updated to $3.0 \times 10^7$ $\mathrm{Matm^{-1}}$ (Nguyen et al., 2014a) which improved model predictions of 2-methyltetrols (supporting information) and total isoprene OA. The diffusivity of isoprene products in the particle ($D_a$, $\mathrm{cm^2 s^{-1}}$ ) was predicted by fitting a line through the data in the work of Song et al. (2015) resulting in

$$D_a = 10^{(7.18RH - 12.7)} \tag{1}$$

for $0 \leq RH \leq 1$. Semisolid aerosol organic aerosol ($D_a < 10^{-12} \ \mathrm{cm^2 s^{-1}}$) is not expected in the humid eastern US during summer (Pajunoja et al., 2016). SOA from later-generation high-$NO_x$/high-$NO_2$ SOA species (methacrylic acid epoxide and hydroxymethyl-methyl-$\alpha$-lactone) is relatively minor (Pye et al., 2013; Marais et al., 2016), consistent with observations from SOAS ground sites (Budisulistiorini et al., 2015). All SOA produced through heterogeneous uptake is assumed nonvolatile in CMAQ v5.1. SOA from isoprene and monoterpene organic nitrates is semivolatile, but the nitrate groups hydrolyze in the particle to produce SOA which is assumed nonvolatile and nitric acid (Pye et al., 2015). SOA from cloud processing is relatively minor in terms of average SOA concentrations. POA and volatility-based SOA is treated as hydrophobic by default, while aqueous and in-cloud SOA is assumed hydrophilic and resides in a water-rich phase (CMAQv5.1 aero6i assumptions, Table 1).

## 2.2 Updating properties of semivolatiles

### 2.2.1 Molecular properties

For SOA systems, the molecular weight and OM/OC by mass must be specified for mass-to-molecule number unit conversions within CMAQ and to allow for post-processing of organic carbon (OC) from total SOA for comparison to observations. The number of carbons per molecule ($n_C$) is also specified for the traditional semivolatile systems to allow for oligomerization to conserve carbon (Carlton et al., 2010). Historically, in CMAQ model formulations (v5.1 and prior), the $n_C$, saturation concentration ($C_i^*$), and OM/OC were set independently with the OM/OC obtained from chamber experiments and $n_C$ set to that of the parent hydrocarbon. The molecular weight was calculated to be consistent with the number of carbons and OM/OC. The OM/OC values were not a function of volatility or peroxy radical ($RO_2$) fate. Thus, all SOA species from a given





parent hydrocarbon were assumed to have the same properties (OM/OC, molecular weight, number of carbons) regardless of their volatility. When viewed in the O:C vs $C_i^*$ space (Baker et al., 2015) this leads to some apparent contradictions such as sesquiterpene SOA being more functionalized and having a longer carbon backbone at a given vapor pressure than isoprene or monoterpene SOA. This inconsistency is also seen in the molecular weight vs. $C_i^*$ space (Figure 4). Most SOA constituents

are expected to show that molecular weight is correlated with vapor pressure ($C_i^*$) with more functionalized species having a shallower slope than less functionalized species (Shiraiwa et al., 2014). Systems examined by Shiraiwa et al. (2014) were found to reside between a line characteristic of O:C=0 (alkane, $C_nH_{2n+2}$) and O:C=1 (sugar, $C_nO_nH_{2n-2}$). Sesquiterpene SOA in CMAQ v5.1 resides outside the molecular corridor bounds that correspond to O:C=0 (OM/OC=1.17) and O:C=1 (OM/OC=2.3 to 2.5). The CMAQv5.1 Odum 2-product isoprene SOA components imply an O:C>1 (which is possible, but not observed by

Shiraiwa et al. (2014) and infrequent in the work of Chen et al. (2015)).

Structure-activity relationships or group contribution methods can be used to relate vapor pressure, molecular weight, and OM/OC (or molar O:C). Donahue et al. (2011) developed a relationship between saturation concentration, number of carbons per molecule, and number of oxygens per molecule ($n_O$) ignoring sulfate and nitrate for use with the 2-D volatility basis set (VBS):

$$\log_{10} C_i^* = 0.475(25 - n_C) - 2.3 n_O + 0.6 n_C n_O / (n_C + n_O) \tag{2}$$

Built into this relationship are assumptions about the functionality of semivolatile organic compounds (specifically equal alcohols and ketones with acid terminal groups), the volatility of a 25 carbon alkane ($C_i^*$=1 µgm$^{-3}$), and how a given functional group affects volatility (from the SIMPOL model (Pankow and Asher, 2008)). Note that considerable variability in atmospheric aging exists in terms of addition of functional groups as indicated on van Krevlen diagrams (Chen et al., 2015). The number of

oxygen is related to the molar O:C by

$$n_O = n_C (O:C) \tag{3}$$

O:C can be related to the mass-based OM/OC (Simon and Bhave, 2012):

$$O:C = \frac{12}{15} \left( \frac{OM}{OC} \right) - \frac{14}{15} \tag{4}$$

Which assumes only H, O, and C atoms and produces results consistent with AMS-determined relationships between O:C

and OM/OC (Canagaratna et al., 2015). OM/OC was the focus of this work instead of O:C since OM/OC values are directly used to post-process model output for comparison to observation network measurements of OC. In addition, OM/OC ratios are a useful quantity in reconstructing total mass of PM and could be available routinely from the Interagency Monitoring of Protected Visual Environments (IMPROVE) network in the future using Fourier transform infrared spectroscopy (FTIR) analysis (Ruthenburg et al., 2014). The molecular weight ($\widetilde{M}$) follows as:

$$\widetilde{M}_i = 12 n_C \left( \frac{OM}{OC} \right) \tag{5}$$

Equations 2 to 5 provide four equations for six unknowns: $n_C$, $n_O$, O:C, OM/OC, $C_i^*$, and $\widetilde{M}_i$. $C_i^*$ were obtained from the Odum 2-product fits (Odum et al., 1996) derived from laboratory data (Carlton et al., 2010; Pye and Pouliot, 2012) and $n_C$




were set to that of the parent hydrocarbon. The OM/OC and molecular weight were then calculated. $n_O$ and O:C were not needed for CMAQ (but could be easily obtained). Pankow et al. (2015) undertook a similar exercise in which they developed surrogates for each of the CMAQ v5.0 SOA species using SIMPOL and plausible structures. Their information was used when available and equations 2 to 5 employed otherwise. For the systems where Pankow et al. (2015) provide information, the results

based on equations 2 to 5 are very similar. For SOA from the explicit later-generation precursors (such as IEPOX, isoprene dinitrates, and monoterpene nitrates) the molecular properties were already tied to a specific surrogate identity. The CMAQ SOA species representing actual compounds were not updated.

### 2.2.2 Deposition properties

The deposition-related properties of gases, such as their solubility, diffusivity, and reactivity, are related to molecular structure

and composition. CMAQ uses a resistance in series method for dry deposition (Pleim and Ran, 2011). CMAQ v4.7 through v5.1 use adipic acid (Henry's law coefficient, $H = 2 \times 10^8 \ \mathrm{Matm^{-1}}$) as a wet deposition surrogate for gas-phase semivolatile organic compounds (SVOCs). Default dry deposition of SVOCs is based on acetic acid (H = $4.1 \times 10^3 \exp(63,000(298 - T)/(298T)) \ \mathrm{Matm^{-1}}$, gas-phase diffusivity ($D_g$) = 0.0944 $\mathrm{cm^2 s^{-1}}$, dry cuticular resistance = 1200 $\mathrm{sm^{-1}}$, Lebas molar volume = 63$\mathrm{cm^3 mol^{-1}}$).

Hodzic et al. (2014) used the Generator of Explicit Chemistry and Kinetics of Organics in the Atmosphere (GECKO) to predict products from various SOA systems commonly represented in models. Henry's law coefficients were then estimated based on the GROup contribution Method for Henry's law Estimate (GROMHE) (Raventos-Duran et al., 2010). GROMHE was found to reproduce Henry's Law coefficients with mean absolute error of about 0.3 log units compared to 0.5 for HenryWin and 0.4 for SPARCv4.2 (Raventos-Duran et al., 2010). For SOA systems, a strong relationship was observed between saturation

concentrations and Henry's law coefficients with chemically aged species being less volatile, more functionalized, and more soluble than their parent hydrocarbon. Although the relationship between H and $C_i^*$ was relatively robust, variability in H spanned many orders of magnitude for a given $C_i^*$ bin. The relationships derived by Hodzic et al. (2014) were used to predict the Henry's law coefficients as a function of $C_i^*$ for each SVOC surrogate in equilibrium with the particle in the model. An enthalpy of solvation of 50 $\mathrm{kJmol^{-1}}$ was also adopted to adjust the Henry's law coefficients for temperature. Note that although

the approach used by Hodzic et al. (2014) is also a group contribution method, it potentially represents the functional groups present in CMAQ SOA species with different groups than would be assumed by equations 2-5.

Additional properties needed for deposition include the gas-phase diffusion coefficient which was calculated as a function of molecular weight via $D_{g,i} = 1.9(\widetilde{M_i})^{-2/3} \ \mathrm{cm^2 s^{-1}}$ (Schnoor, 1996) and the LeBas molar volume ($V_{LeBas}$) calculated assuming ring-opened products (Mackay et al., 1993):

$$V_{LeBas} = 14.8n_C + 7.4n_O + 3.7n_H \mathrm{cm^3 mol^{-1}} \tag{6}$$

where the number of hydrogens, $n_H$, is calculated from the molecular weight assuming only carbon, oxygen, and hydrogen. Modifications were also made to the deposition parameters affecting $H_2O_2$, IEPOX, and organic nitrates to produce results consistent with Nguyen et al. (2015a) (parameters available in the supporting information).





## 2.3 Predicting phase separation

The solubility of an organic compound in water generally decreases due to addition of a salt with some exceptions like glyoxal (Kampf et al., 2013). However, as atmospheric aerosols contain water, salts, and organic compounds, there are likely conditions where the solubility of an organic is more or less favorable in the water/inorganic-rich phase. Mixed organic-inorganic solutions

have been observed to phase separate into an organic-rich and inorganic-rich phase based on their degree of functionalization (as measured by O:C) and relative humidity. The O:C serves as a proxy for molar polarization which dictates the magnitude of salting out effect through the Setchenov equation (Bertram et al., 2011). The relative humidity above which a single combined phase exists is called the separation relative humidity (SRH). In CMAQ, when the ambient relative humidity was below the SRH, the model separated the particle into a water-rich phase (containing aqueous SOA) and organic-rich phase (containing

traditional SOA and POA). This separation of aqueous SOA and traditional SOA at low RH is consistent with the work of Ye et al. (2016) who found that isoprene surrogates unfavorably interacted with $\alpha$-pinene SOA even at 60% RH. The SRH is not expected to be a strong function of the organic-to-inorganic ratio (Bertram et al., 2011; You et al., 2013), molecular weight of the organic species, or temperature between 244 and 290K (You and Bertram, 2015). However, the SRH is a function of the type of salt present with ammonium sulfate having higher SRH than ammonium bisulfate, sodium chloride, and ammonium nitrate

for a given O:C. During SOAS, inorganic aerosol was dominated by $(NH_4)_2SO_4$ and $NH_4HSO_4$, and SRH was diagnosed in CMAQ based on the You et al. (2013) experimental results for ammonium sulfate. The relationship for SRH (fraction between 0 and 1) as a function of O:C was recast in terms of OM/OC:

$$SRH = \left[ 1 + \exp\left( 7.7\frac{OM}{OC} - 15.8 \right) \right]^{-1} \tag{7}$$

Since ammonium sulfate has the highest SRH of the salts examined by You et al. (2013), choosing another salt would increase

the frequency of phase-mixing and difference compared to the base simulation.

## 2.4 Predicting water uptake to the organic phase

Water uptake to the organic phase was predicted using $\kappa$-Köhler theory and solving for the volume equivalent diameter, D (Petters and Kreidenweis, 2007):

$$RH - \frac{D^3 - D_{core}^3}{D^3 - D_{core}^3(1-\kappa)} \exp\left( \frac{4\sigma_w \widetilde{M}_w}{RT\rho_w D} \right) = 0 \tag{8}$$

where $D_{core}$ is the volume equivalent accumulation mode diameter excluding water associated with organic species, $\widetilde{M}_w$ is the molecular weight of water, $\rho_w$ is the density of water, $R$ is the universal gas constant, $T$ is temperature, and $\sigma_w$ is the surface tension of water (0.072 J m$^{-2}$). In order to calculate the volume equivalent diameters, $D$ and $D_{core}$, particle density was needed. Density values in CMAQ v4.7-v5.1 for organic constituents are generally on the order of 2,000 kgm$^{-3}$. The densities of organic aerosol species were updated to chamber specific information when available (Ng et al., 2007; Chan et al., 2009)

and 1,400 kgm$^{-3}$ otherwise. The mass of particle liquid water associated with organic compounds per volume of air (W$_o$) was





calculated from:

$$W_o = \frac{\pi N_p \rho_w}{6}(D^3 - D_{core}^3) \tag{9}$$

where $N_p$ was number of particles per volume air. Total aerosol water in the model was computed as the sum of water associated with inorganics ($W_i$) calculated with ISORROPIA v2.2 (Fountoukis and Nenes, 2007) and $W_o$.

The hygroscopicity parameter, $\kappa$, was calculated as a volume weighted sum of the individual component $\kappa_i$ (Petters and Kreidenweis, 2007):

$$\kappa = \sum_{i=1}^{n} \frac{\kappa_i V_{core,i}}{V_{core}} \tag{10}$$

CCN-based $\kappa$s were used following Lambe et al. (2011) due to the completeness of that study. The O:C values obtained by Lambe et al. (2011) were increased 27% to account for a low bias in old calibrations (Canagaratna et al., 2015). In addition, the relationship was recast in terms of OM/OC resulting in:

$$\kappa_{org,i} = 0.11\frac{OM}{OC} - 0.10 \tag{11}$$

Equations in terms of O:C are available in the supporting information.

For subsaturated conditions, like those relevant to predicting water uptake, the hygroscopic growth factor (hgf) $\kappa$ is most relevant (Pajunoja et al., 2015); however, simulations used CCN-based $\kappa_{org,i}$ to predict water uptake. Hgf-based kappas from Duplissy et al. (2011) and Raatikainen et al. (2010) were combined with data from Jimenez et al. (2009) into a parametrization by Lambe et al. (2011). After correcting the parameterization to use updated O:C, the parametrization including hgf-based data resulted in one negative kappa and three kappas higher than 0.6 (same as ammonium sulfate), which may be an upper limit on $\kappa_{org,i}$ (Ervens et al., 2011). Thus, contrary to the typical trend of $\kappa_{CCN} > \kappa_{hgf}$, more than half of the species had $\kappa_{CCN} < \kappa_{hgf}$. Variation from study to study may be higher than $\kappa_{CCN}$ vs $\kappa_{hgf}$ variations, which have been found to be within 30% for many compounds and unable to be resolved using common measurement techniques (Petters and Kreidenweis, 2007).

In processing of model output, the following equation was used to determine how errors in the concentration of organic compounds, $\kappa_{org}$, and RH propagated to errors in $W_o$:

$$W_o = \frac{\rho_w}{\rho_{org}}[OA]\kappa_{org}\frac{1}{(1/a_w - 1)} \tag{12}$$

with the activity of water ($a_w$) defined as

$$a_w = \frac{RH}{\exp\frac{4\sigma_w \widetilde{M}_w}{RT\rho_w D}}. \tag{13}$$

## 2.5 Representing the effect of water on semivolatile partitioning

Partitioning of semivolatile organic species into an absorbing medium can be described by Raoult's law:

$$\frac{A_i/M_p}{G_i} = \frac{RT}{\widetilde{M}_p \gamma_i P_i^{sat}} \tag{14}$$





where $A_i$ is the aerosol-phase concentration of species i ($\mu$gm$^{-3}$air), $G_i$ is the gas-phase concentration of i ($\mu$gm$^{-3}$air), $M_p$ is the mass of the partitioning medium ($\mu$gm$^{-3}$air), $\widetilde{M_p}$ is the molecular weight of the partitioning medium, $\gamma_i$ is a mole-based activity coefficient, and $P_i^{sat}$ is the saturation vapor pressure of pure i. This relationship (equation 14) is true regardless of how the partitioning coefficient ($C_i^*$ or $K_{p,i}$) is defined. CMAQ, following Schell et al. (2001), defines $C_i^*$ as:

$$C_i^* \equiv \frac{\widetilde{M_i}\gamma_i P_i^{sat}}{RT} \tag{15}$$

where the relevant molecular weight is the individual species molecular weight in contrast to the traditional definition of Pankow (1994) that uses the partitioning medium's molecular weight:

$$C_i^{*'} = \frac{1}{K_{p,i}} \equiv \frac{\widetilde{M_p}\gamma_i P_i^{sat}}{RT} \tag{16}$$

Model calculations in this work used the definition in equation 15 thus:

$$C_i^* = \frac{G_i \widetilde{M_i} N}{A_i} \tag{17}$$

where the total moles in the partitioning medium (N) is

$$N = N_{other} + \sum_i (A_i/\widetilde{M_i}) \tag{18}$$

$N_{other}$ represents aerosol in the partitioning medium that is not semivolatile during calculation. Including water in the partitioning medium (either from uptake onto hydrophilic organic compounds or from the inorganic phase) increases the moles of partitioning medium by contributing to $N_{other}$. Inclusion of water, and even inorganic constituents, in the absorbing phase has been encouraged for simplified models in order to reproduce more detailed calculations (Zuend et al., 2010).

One equation for one unknown can be derived where $T_i$ is the total ($G_i + A_i$) mass of the semivolatile determined by the mass-based stoichiometric coefficients and amount of parent hydrocarbon reacted ($\alpha_i \Delta HC$):

$$f(N) = 0 = \frac{N_{other}}{N} - 1 + \sum_i \frac{T_i}{C_i^* + \widetilde{M_i}N} \tag{19}$$

Equation 19 was solved for $N$ in the model.

$\widetilde{M_p} \approx \widetilde{M_i}$ for interpretation of data from chamber experiments only, and it allows for $C_i^{*'} \approx C_i^*$ in a single-precursor chamber experiment so that the Odum 2-product fit can be determined. Table 2 indicates this was a realistic assumption for most systems as the two surrogate molecular weights vary by less than 10%. This assumption was not necessary within the CMAQ model.

## 2.6 Estimating solubility and deviations from ideality

To represent deviations from ideality, the saturation concentration used in Raoult's law was adjusted using an activity coefficient. All organic-organic interactions were assumed ideal and only the inclusion of water drove deviation from ideality.



Observations during SOAS indicate that despite a factor of seven change in ambient aerosol water concentration from night to day, $x_w$ (mole fraction water in the partitioning medium) typically varied over a narrow range (80% to 96% by mole) throughout the day. The activity coefficient for each organic species, $\gamma_i$, was determined using a one-constant Margules equation:

$$\ln(\gamma_i) = x_w^2 \ln(\gamma_i^\infty) \tag{20}$$

Since $\gamma_i^\infty$ (the temperature dependent constant in the Margules equation) corresponds to the activity coefficient at infinite dilution in water ($x_w = 1$), it can be estimated based on Henry's law combined with Raoult's law:

$$\gamma_i^\infty = \frac{\widetilde{M_i}\rho_w}{H_i C_{0,i}^* RT \widetilde{M_w}} \tag{21}$$

where $C_{0,i}^*$ is the pure species saturation concentration at T. $\gamma_i^\infty$ is related to solubility ($S_i$) in mass per volume of water:

$$S_i = H_i C_{0,i}^* RT \tag{22}$$

The saturation concentration as a function of water becomes

$$C_i^* = C_{0,i}^* (\gamma_i^\infty)^{N_w^2/N^2} \tag{23}$$

where $N_w$ is the moles of aerosol water in the partitioning medium. This equation applies across the entire organic to water spectrum and shows that $\gamma_i^\infty$ represents $C_i^*$ of a species in water ($x_w = 1$) normalized to the pure species $C_{0,i}^*$. Evaluating $C_i^*$ for pure water provides, $C_{H,i}^*$, the saturation concentration at infinite diulution:

$$C_{H,i}^* = \frac{\widetilde{M_i}\rho_w}{H_i RT \widetilde{M_w}} \tag{24}$$

Values are available in Table 2. The solubilities of nonvolatile species derived from traditional precursors (oligomers/accretion products) were estimated based on assuming a $C_{0,i}^*$ between $10^{-2}$ and $10^{-5}$ µgm$^{-3}$ and the Henry's law coefficients of Hodzic et al. (2014).

This representation of deviations from ideality resulted in competing effects due to the addition of aerosol water to the partitioning medium. Adding water increased the partitioning medium as described in section 2.5, which led to more SOA. However, adding water also increased the activity coefficient via the Margules model (Figure S2) leading to higher $C_i^*$ and less favorable partitioning (Figure S3). The Margules model, combined with the fact that all deviations are observed to be positive for the species examined here indicated large additions of water reduced SOA due to the activity coefficient adjustment. Indeed, all saturation concentrations for partitioning into pure water ($C_{H,i}^*$) are higher than those into pure organic ($C_{0,i}^*$) by one to four orders of magnitude (Table 2). *A priori* assumptions regarding the solubility and activity of monoterpene nitrates were so non-ideal that particulate nitrate was driven entirely out of the particle, inconsistent with observations (Xu et al., 2015a, b). As a result, the Henry's law coefficient for monoterpene nitrates (MTNO$_3$) was increased by a factor of 100 and all activity coefficients were reduced by a factor of 10 compared to *a priori* values in the CMAQ $\gamma \neq 1$ simulation (detailed in a subsequent section). These adjustments may have been necessary due to inaccuracies in the Henry's law coefficients, pure species saturation concentrations, molecular weights, Margules model, or a combination of all of the above.





### 2.7 Estimating WSOC

WSOC is an operationally defined species measured by adding water to a system and analyzing the dissolved compounds (Sullivan et al., 2004). Particulate compounds with solubilities greater than 10 g/L tend to be measured as WSOC regardless of the sampling and extraction method while compounds with solubilities less than $1 \times 10^{-4}$ g/L are insoluble (Psichoudaki and Pandis, 2013). To determine the fraction of OA extracted as WSOC ($\text{WSOC}_p$), the particle phase can be modeled as an equilibrium between two phases: a and b. The fraction of species, i, in phase a compared the total particulate species concentration is:

$$f_{a,i} = \left(1 + \frac{C_{a,i}^* N_b}{C_{b,i}^* N_a}\right)^{-1} \tag{25}$$

where $N_a$ and $N_b$ are the number of moles in phases a and b respectively. If phase b has no water and is ideal while phase a is dominated by water and obeys Henry's law, then the fraction of aerosol species i extracted as $\text{WSOC}_p$ ($f_{WSOC,i}$) is

$$f_{WSOC,i} = \left(1 + \gamma_i^\infty \frac{WIOA}{LWC} \frac{\widetilde{M_w}}{\widetilde{M_i}}\right)^{-1} \tag{26}$$

where WIOA and LWC are concentration of water insoluble OA and liquid water in mass per volume of air. Thus WSOC depends on the amount of insoluble material, liquid water, Henry's law coefficient, and pure species saturation concentration.

### 2.8 Simulations

CMAQ v5.1 (Appel et al., in preparation) with additional updates was run over the eastern United States for June 2013 at 12km by 12km horizontal resolution using the same domain and meteorological inputs as in the work of Pye et al. (2015). Anthropogenic emissions were based on the EPA National Emission Inventory (NEI) 2011 v1. Isoprene emissions were predicted with the Biogenic Emission Inventory System (BEIS) v3.6.1 (Bash et al., 2016). BEIS often predicts lower emissions than the Model of Emissions of Gases and Aerosols from Nature (MEGAN) (Carlton and Baker, 2011), and isoprene emissions were increased by 50% in this work to better agree with observations of isoprene and OH at the SOAS Centreville, AL (CTR) site (Figure S1i-h).

A baseline simulation including surrogate property updates detailed in section 2.2 (molecular weight, Henry's law coefficients, etc) and three sensitivity simulations examining the implications of aerosol liquid water for SOA were conducted (Figure 3). In the baseline simulation, POA and traditional SOA were designated hydrophobic and did not interact with aerosol water or SOA produced through aqueous pathways following common chemical transport model assumptions. Two sensitivity simulations examined the implications of aerosol water on semivolatile partitioning via increases in the partitioning medium assuming ideal mixing. In one simulation, POA, traditional SOA, aqueous SOA, and water associated with inorganic constituents were assumed to form one ideal phase when RH was above the separation relative humidity (SRH) and undergo liquid-liquid phase separation (LLPS) into organic-rich (POA and traditional SOA) and water-rich (aqueous SOA and inorganic constituents) ideal phases otherwise. In the second simulation, uptake of water to the organic phase ($W_o > 0$) was predicted based on its OM/OC





and $\kappa$-Köhler theory (Petters and Kreidenweis, 2007). The impacts of phase separation and water uptake to organic species along with deviations from ideality via an activity coefficient ($\gamma$) were simulated together in the third sensitivity simulation ($\gamma \neq 1$). *A posteriori* parameters used in $\gamma \neq 1$ are available in Table S6.

### 2.9 Observations for Evaluation

Simulations were evaluated by comparing to OC from IMPROVE, Chemical Speciation Network (CSN), and SouthEastern Aerosol Research and Characterization (SEARCH) network observations in the Eastern U.S. For comparisons to SEARCH observations, the Jefferson Street, Atlanta, GA (JST) and Birmingham, AL (BHM) urban sites as well as Yorkville, GA (YRK) and Centreville, AL (CTR) rural sites were considered. In order to estimate secondary organic carbon (SOC), the method of Yu et al. (2007), which uses OC/EC ratios, was revised to account for the semivolatile nature of POA. The total nonvolatile

POA in CMAQ is assumed to correspond to emissions of $C_i^* \approx 3000\ \mu\mathrm{gm}^{-3}$ and lower volatility compounds. The volatility distribution of gasoline vehicle POA from May et al. (2013) and used by the CMAQ-VBS (Koo et al., 2014) was used to estimate how much POA is expected in the particle under ambient conditions.

The fraction of POA in the particle ($f_P$) for each observation data point was estimated as:

$$f_p = \sum_{i=1}^{5} \frac{\alpha_i}{1 + C_i^*/(OC_{obs}(OM/OC)_{mod})} \tag{27}$$

where the volatility profile is described by $C_i^*$ of 0, 1, 10, 100, and 1000 $\mu\mathrm{gm}^{-3}$ species in the following mass-based abundance: 0.27, 0.15, 0.26, 0.16, and 0.17. Observed SOC was estimated from each observed OC by

$$SOC_{obs} = OC_{obs} - f_p(POC/EC)_{mod}EC_{obs} \tag{28}$$

therefore

$$POC_{obs} = f_p(POC/EC)_{mod}EC_{obs} \tag{29}$$

This calculation only accounts for the effect of dilution and partitioning on POC and does not account for chemical processing that may convert POA to SOA. In addition, compared to other volatility profiles such as diesel POA, this profile tends to be weighted toward lower volatility compounds. As a result, this approach may be an upper bound on the amount of POC (lower bound on SOC).

In addition to the routine monitoring network data, model predicitons were compared to data from the Centreville, AL

(CTR, 87.25° W longitude, 32.90° N latitude) and Look Rock, TN (LRK, 83.94° W longitude, 35.63° N latitude) sites from the SOAS field campaign in the southeast United States. Observations include water-soluble organic carbon in both particle and gas phase (Xu et al., in preparation; Sullivan et al., 2004), aerosol LWC (Nguyen et al., 2014b; Guo et al., 2015), OA (Xu et al., 2015a, b; Budisulistiorini et al., 2015; Hu et al., 2015), and gas-phase species (Nguyen et al., 2015a; Budisulistiorini et al., 2015). The supporting information provides additional evaluation such as a comparison to OH (Feiner et al., submitted),

isoprene (Su et al., 2016; Misztal et al., in preparation), and 2-methyltetrol (Isaacman et al., 2014) concentrations.



## 3 Results and Discussion

### 3.1 Updated Base Model

#### 3.1.1 Effect of Property Updates

Figure 4 shows the updated molecular weights as a function of saturation concentration and colored by OM/OC. Values are
summarized in Table 2. Four species that were initially outside the O:C=0 and O:C=1 bounds in CMAQ v5.1, AISO1, AISO2, ASQT and, ABNZ3, were moved within the bounds or just slightly outside as a result of implementing equations 2 through 5 for traditional OA. The impact of updated OM/OC and molecular weight had small impacts on OM (up to 4% decreases) and larger impacts on OC (5-8% decrease in OC across the Southeast). This change was driven by an increase in the OM/OC of biogenic (semivolatile isoprene and monoterpene) SOA.

Note that there is likely inconsistency in the structure and assumed vapor pressure for 2-methyltetrols and 2-methylglyceric acid. The model considers IEPOX-derived SOA to be mainly 2-methyltetrols and organosulfates with a small amount of oligomers (Pye et al., 2013). All IEPOX-derived species were treated as nonvolatile, but they should be semivolatile given their molecular weight. Lopez-Hilfiker et al. (2016) indicate that IEPOX-derived organosulfates and 2-methyltetrols measured by common techniques include decomposition products of accretion reactions and that IEPOX-SOA should be relatively
nonvolatile, consistent with Hu et al. (2016) and the nonvolatile assumption here. The nonvolatile assumption is, however, inconsistent with 2-methyltetrols being present in the gas-phase as observed by Xie et al. (2014). The glyoxal SOA in CMAQ also corresponded to a monomeric unit. If oligomers are the dominant form for aqueous methylglyoxal SOA (Altieri et al., 2008), then the molecular weight would need to be increased. Given the nonvolatile nature of IEPOX-derived SOA and glyoxal SOA, they were not significantly affected by the sensitivity simulations.

In the base and updated models, dry deposition of OA played a relatively minor role in removing semivolatile compounds from the system. Volatility was the primary factor determining the relative role of gas vs particle deposition for a given species with the specific value of the Henry's law coefficient being less important as indicated by relatively small changes in overall deposition between the base and update (Figure 5). At 298K, the less volatile SVOCs became more soluble than predicted by base CMAQv5.1 while the more volatile SVOCs became less soluble. With the new parameters, dry deposition of gas-phase
SVOCs increased 20% while wet deposition decreased 6%. Total SOA+SVOC deposition changed by less than 2% and surface concentrations changed by less than 3%. Overall, particle-phase deposition accounted for 22% of the loss of SOA+SVOC mass. Dry deposition of gas-phase SVOCs accounted for 32%, and wet deposition of gas-phase SVOCs accounted for 46%. The relative role of gas-phase SVOC wet deposition was twice as important as predicted by Hodzic et al. (2014), consistent with the greater contribution from soluble biogenic species in the southeast in this work. The combined effects of the molecular
weight, OM/OC, and deposition updates resulted in a 10% decrease in predicted OC over the Southeast.



### 3.1.2 Isoprene SOA

Heterogeneously-derived IEPOX SOA in CMAQ was assumed to be nonvolatile, and thus, was not greatly affected by the sensitivity simulations. Positive matrix factorization (PMF) analysis of Aerosol Chemical Speciation Monitor (ACSM) data and high resolution time-of-flight aerosol mass spectrometer (HR-ToF-AMS) data identified a factor with prominent $m/z$ 82

signals (Lin et al., 2012; Xu et al., 2015a, b). This factor was referred to as "IEPOX-OA" and "isoprene-OA", respectively. While it is largely attributed to IEPOX uptake, it may not entirely due to IEPOX (Xu et al., 2015a, b; Schwantes et al., 2015). The term "isoprene-OA" will be used to refer to the ambient PMF factor regardless of technique.

Liu et al. (2015) report that only half of the isoprene $RO_2 + HO_2$ SOA is from IEPOX in laboratory experiments. Furthermore, the AMS isoprene-OA PMF factor is not fully speciated. During SOAS at the CTR site, Lopez-Hilfiker et al. (2016) were

able to explain roughly 50% of the AMS isoprene-OA at the molecular level. Hu et al. (2015) explained 78% of isoprene-OA at CTR by molecular tracers measured on-line (Isaacman et al., 2014) and identified on filters, but only 26% of isoprene-OA was linked to tracers at LRK (Budisulistiorini et al., 2015). The lack of mass closure in these studies may have resulted from a lack of authentic standards for quantifying accretion products (oligomers and organosulfates).

Regional modeling also indicates a number of later-generation species besides IEPOX contribute significantly to isoprene-

derived SOA in the United States. Marais et al. (2016) indicate that isoprene SOA in the eastern US consists mainly of IEPOX (58%) and glyoxal (28%) uptake products with 14% due to other species. Ying et al. (2015) attribute only 20% of isoprene-OA to IEPOX uptake with roughly an equal contribution from MGLY uptake. Semivolatile isoprene OA and its oligomers accounted for just under 10% of isoprene OA in their work. Thus, it is unclear if models can consider only SOA from IEPOX for the isoprene system as a surrogate for AMS measured isoprene-OA.

Figure 6 shows three model definitions of isoprene-OA: SOA due only to IEPOX reactive uptake, SOA due to IEPOX reactive uptake and semivolatile isoprene+OH products, and SOA due to IEPOX and glyoxal/methylglyoxal uptake. Also included are the PMF factor observations of isoprene-OA from Xu et al. (2015a) for CTR and Budisulistiorini et al. (2015) for LRK. SOA is examined relative to sulfate as sulfate provides the acidity and aerosol medium for heterogeneous uptake (Pye et al., 2013; Marais et al., 2016). Modeled SOA due to IEPOX reactive uptake was increased relative to CMAQ v5.1 as a result of the higher

rate constant for organosulfate formation implemented in this work compared to the work of Pye et al. (2013). At the CTR site, all definitions of isoprene OA led to overestimates of observed isoprene-OA relative to sulfate. Isoprene OA based on IEPOX uptake + semivolatile Odum 2-product surrogates led to the highest predicted concentrations and a slope of 0.70 compared to the observed slope of 0.45. As a result, CMAQ IEPOX-OA could respond more strongly to changes in $SO_x$ emissions than ambient data would suggest as the regression coefficient has been interpreted as the magnitude of the sulfate control on isoprene

OA (Xu et al., 2015a). However, no direct relationship between Odum 2-product isoprene SOA and sulfate exists in CMAQ. The correlation between isoprene OA and sulfate for all three model representations was high (r>0.8) and close to the observed value (r=0.91), which is also consistent with ongoing modeling work with CMAQv5.1 (Vasilakos et al., in preparation). On an absolute basis, predicted IEPOX+SV OA reproduced observed isoprene OA within 6% overall with small underestimates in the afternoon. However, sulfate concentrations were low by 30% and ISOPOOH+IEPOX concentrations (Nguyen et al., 2015a)





were overestimated by a factor of 2.4 in the model consistent with other modeling work (Vasilakos et al., in preparation). Thus, as better agreement is obtained for the gas-phase isoprene species, additional increases in processes that convert isoprene $RO_2 + HO_2$ products to the particle phase may be needed despite the overestimates relative to sulfate shown in Figure 6 for Centreville. These additional processes may include accounting for partitioning of 2-methyltetrols to the gas phase, formation

of $C_5$-alkene triols, and/or faster oligomerization (Lopez-Hilfiker et al., 2016; Xie et al., 2014; Surratt et al., 2010).

At LRK, the different model representations of isoprene-OA closely resembled the observed isoprene-OA to sulfate ratio of 0.83. IEPOX uptake alone resulted in a slope of 0.61 and IEPOX uptake along with semivolatile isoprene+OH products result in a slope of 0.79. The model showed only a slightly stronger relationship to sulfate than the observations (observed r=0.87) with the different model representations indistinguishable in their correlation with sulfate (model r=0.93 to 0.95). Similar to the

10 model at the CTR site, CMAQ underpredicted sulfate at LRK by about 30%. IEPOX+SV isoprene OA was biased low by almost 40% and the bias in isoprene-OA (IEPOX+SV) was correlated with the bias in sulfate. ISOPOOH+IEPOX was underestimated by 60% at LRK, in contrast to the CTR site where it was overestimated (supporting information). Thus, isoprene products in the model were too efficiently converted to SOA at LRK despite the low sulfate.

### 3.1.3 Total OA

Model predictions of OC, SOC, and POC were compared to network observations using the methods described in section 2.9 to determine how model errors in POA (specifically the nonvolatile assumption) could mask errors in SOA. An IMPROVE network observation with a value of $16.9\,\mu g Cm^{-3}$ (at SHMI1, Shamrock Mine, CO) had a Cook's distance (Cook, 1977) much greater than 1 in a base model-observation comparison and was subsequently removed from all analysis. For the IMPROVE network, 86% of observed OC was predicted to be secondary in nature (equation 28) while CMAQ predicted 46% of OC was

secondary. The standard deviation (s) in model predicted SOA fraction was much higher at 0.21 vs 0.08 in observations. The CSN network (with a greater proportion of urban sites) was slightly less secondary in nature with 79% of OC as SOC (s = 0.11) and CMAQ predicting 40% of model OC as SOC (s = 0.19). The SEARCH network was the most influenced by SOA of the three networks. SEARCH OC was predicted to be 88% SOC (s = 0.06) while CMAQ indicated 58% SOC (s = 0.19). PMF analysis at the urban JST site during summer 2011 and 2013 indicates that POA (HOA, BBOA, COA) factors accounted for

18-30% of total OA (Xu et al., 2015a; Budisulistiorini et al., 2013), while CMAQ predicted a 42% contribution of POC to OC averaged across the urban and rural sites here.

Figure 7 indicates that overestimates in POC roughly compensated for underestimates in SOC in the updated CMAQ model. CMAQ predicted total OC was within 20% of average observed OC across each network. The normalized mean bias (NMB) for POC and SOC was much larger in magnitude than for total OC but relatively constant between networks. Specifically,

SOC was low by 40% while POC was high by a factor of 1.7 to 1.8. The overestimate in model POC at the routine network locations was consistent with the model overestimate in AMS/ACSM-measured POA at SOAS CTR and LRK sites. Neither site resolved a hydrocarbon-like (HOA) type aerosol (Xu et al., 2015a; Budisulistiorini et al., 2015), indicating that POA from fossil fuel sources contributed less than 5% of total OA. A biomass burning (BBOA) type aerosol was resolved at the CTR





site and episodic in nature. Comparing CMAQ predicted POA from all sources to the BBOA factor at CTR indicated CMAQ generally overestimated POA by a factor of 2, similar to the overestimate for network OC observations.

Additional insight into biases can be obtained by examing the diurnal profiles of OC (Figure 8). The diurnal profile of observed OC is relatively flat at the SEARCH sites, consistent with flat total OA (Xu et al., 2015b). CMAQ predictions had a
pronounced diurnal profile with higher concentrations (and relatively good performance or overpredictions) at night and lower concentrations (coinciding with underestimates) during the day. Averaged across the two urban sites (JST and BHM), however, CMAQ showed no bias as a result of compensating diurnal and spatial errors. Rural OC (YRK and CTR) was underpredicted by about one third. Also included in Figure 8 is the diurnal profile of primary organic carbon (POC) in red dashes. Modeled POC at the Atlanta site correctly showed high concentrations in the morning (6am) and evening (7pm), but tended to peak
several hours earlier than hydrocarbon-like organic aerosol (HOA) observed at JST in 2012 (Budisulistiorini et al., 2016). JST total model-predicted POC during morning and evening transition hours was roughly the same magnitude as total observed OC, further indicating CMAQ tends to overestimate primary organic aerosol.

## 3.2    Role of water

### 3.2.1    Effect on network OC

Figure 9 shows how including water interactions in absorptive partitioning calculations affected model performance compared to routine monitoring networks. While including water associated with inorganic species (LLPS simulation) in the partitioning medium for SOA decreased the bias in SOC for all networks, it led to small increases in the mean error. Except for the SEARCH network, including organic water ($W_o > 0$) also reduced the mean bias at the expense of mean error. The simulation taking into account nonideality ($\gamma \neq 1$) resulted in low normalized mean bias ($\leq 10\%$) and large improvements in the mean bias compared
to all other simulations. The mean error for $\gamma \neq 1$ was marginally increased over the base simulation.

Figure 9 highlights that increases in bias occured during the night (SEARCH network). The largest increases in bias occurred for the $W_o > 0$ simulation as a result of a large contribution of organic water. Similar to the results for the CTR site (Section 3.2.3), daytime concentrations of SOC increased, but were still low compared to observations. In general, the variability in the bias increased as a result of water interactions while the mean bias decreased.

Some caution should be applied when comparing model predictions and observations. Measurements of total aerosol mass from IMPROVE and CSN networks are made under relative humidities of 30-50% and quartz filters for OC analysis from IMPROVE may be subject to ambient conditions in the field and during shipping before analysis (Solomon et al., 2014). Exposure to low RH could cause evaporation of reversible aqueous SOA (El-Sayed et al., 2016). Kim et al. (2015) have indicated the IMPROVE measurements of OC were 27% lower than colocated SEARCH measurements during the summer of
2013 due to sampling artifacts. Episodic field campaign observations may be subject to sampling biases as well. Generally, all aerosol water is expected to evaporate in an aerodynamic lens inlet used on many instruments (Zelenyuk et al., 2006), which can cause changes in the aerosol phase state (Pajunoja et al., 2016) and could potentially lead to changes in partitioning of soluble organic compounds.



### 3.2.2 Frequency of phase separation

Figure 10a shows the June 2013 predicted average OM/OC across the model domain for the simulation in which phase separation was predicted (LLPS). Emitted POA in CMAQ has an OM/OC of 1.25 for vehicles, 1.7 for biomass burning, and 1.4 for other sources, and heterogeneous aging of the POA results in the OM/OC increasing with time (Simon and Bhave, 2012). The

urban sites of Birmingham, AL and Atlanta, GA had predicted OM/OC ratios between 1.3 and 2.2 with a mean of 1.8, while the rural SEARCH sites of Centreville, AL and Yorkville, GA had values between 1.7 and 2.2 with a mean of 1.9, consistent with previous work (Simon et al., 2011).

You et al. (2013) found that particles never undergo phase separation for OM/OC above 2.2 (O:C=0.8) and are always phase separated when OM/OC is less than 1.8 (O:C=0.5). Based on Figure 10b and equation 7, phase separation was a frequent, but

not constant, occurrence. Phase separation was predicted to be more common in urban areas where OM/OC was low and near the Western portion of the domain where RH was low. Figure 10c shows RH, SRH, and phase separation for the CTR site. During the day, SRH decreased as a result of increasing OM/OC ratios for both SOA and POA. The increase in frequency of separation during the day was driven by low RH values during the day. At CTR, the highest frequency of phase separation occurred in the late morning. For other sites, separation was more frequent in the afternoon. These results demonstrate the

complexity of aerosol phase behavior in the atmosphere, and this complexity impacts the way observations are collected and interpreted.

Model predicted RH was low compared to the observed RH by about 6% (mean bias). Since phase separation occurred when RH was below the SRH, the frequency of separation using model RH was biased high. In addition, since the model used the SRH predicted for ammonium sulfate, predictions further represented an upper bound on the frequency of phase separation.

Thus, particles should be internally mixed without phase separation more often than reported in this work. As phase separation was most consistent with default model assumptions, parameterizing the SRH using data from another salt (and using observed RH) would only increase OA as a result of a greater frequency of inorganic water in the partitioning medium. The LLPS simulation represents a lower bound on the effect of inorganic water on the partitioning medium for OA.

### 3.2.3 Effect of water on OA concentrations at CTR

Figure 11 shows the influence of water on aerosols at the Centreville SOAS site during June 2013. The base simulation underestimated OA overall, but most substantially during the day. Including inorganic water in the partitioning medium when RH>SRH (LLPS simulation) resulted in increased OA concentrations at all times of day. Reducing phase separation (as in LLPS compared to Base) has been shown to increase OA concentrations in box modeling (Topping et al., 2013). In CMAQ, concentrations of OA predicted in LLPS were 1.5 times higher than observations at night when RH and aerosol liquid water

concentrations were highest. Note that nocturnal mixing may be underestimated in the model as indicated by low boundary layer depths, high monoterpene concentrations, and high $NO_x$ concentrations compared to observations at night (Pye et al., 2015). The simulation considering uptake of water onto the organic phase ($W_o > 0$) produced the highest predicted OA concentrations out of all simulations as a result of feedback in the model. Specifically, uptake of water and inclusion in the partitioning





medium caused OA concentrations to increase, which further increased the amount of water in the particle and OA. Daytime OA predictions did not exceed observations, but nighttime model concentrations were a factor of 2 higher than observed. A comparison of model-predicted aerosol water with observed aerosol water (Figure 11e) indicated that the model over-predicted aerosol LWC by 2-3x at night when interactions were ideal in the $W_o > 0$ simulation.

The simulation accounting for nonideality in addition to phase separation and uptake of water onto organic compounds ($\gamma \neq 1$), produced results similar to the simulation considering ideal interactions with inorganic water (LLPS) in terms of total OA as a function of time of day (model:observation correlation coefficient=0.5, NMB=10% ($\gamma \neq 1$), 20% (LLPS)). However, the composition of the aerosol was different. Both simulations in which water interactions were ideal (LLPS and $W_o > 0$) resulted in over-predictions of less oxidized oxygenated aerosol (LO-OOA) and particle-phase organic nitrates (supporting

information). Even with the factor of 100 increase in Henry's law coefficient for monoterpene nitrates and factor of 10 decrease in activity coefficient implemented in $\gamma \neq 1$ compared to *a priori* estimates, the predicted concentration of organic-nitrate derived SOA did not substantially change between the base and $\gamma \neq 1$ simulation. The nonideality resulting from including water roughly compensated for the increase in partitioning medium in the case of organic nitrates.

### 3.2.4   Predicting water uptake onto organic compounds

All simulations indicated OM/OC ratios tend to peak during the day and were near a value of 2, consistent with observations (Figure 11c). Semivolatile SOA in the model tended to have lower OM/OC ratios than nonvolatile SOA, which resulted in lower OM/OC ratios overall in the sensitivity simulations compared to the base. These differences in OM/OC between the simulations propagated to predicted $\kappa_{org}$ values (Figure 11d). The base simulation best agreed with the observationally constrained $\kappa_{org}$ values of Cerully et al. (2015), but the model $\kappa_{org}$ was biased low in all simulations.

Basing the $\kappa$ values for organic species on OM/OC (or O:C) may tend to overestimate the $\kappa$ values for organic nitrates (Suda et al., 2014). However, good agreement with the LO-OOA factor ($\kappa$=0.08±0.02, (Cerully et al., 2015)) is obtained for a 50/50 mixture of $MTNO_3$ and it's hydrolysis product ($\kappa$= 0.09). The predicted monoterpene SOA $\kappa$ (0.1) is in agreement with laboratory values ($\kappa$=0.03 to 0.14 (Alfarra et al., 2013)). In addition, the $\kappa$ for monoterpenes is higher than the $\kappa$ for sesquiterpenes consistent with the trend (but not magnitude) in the work of Alfarra et al. (2013). The $\kappa$ for IEPOX-derived OA

(Table 2) was consistent with Isoprene-OA value of of Cerully et al. (2015) ($\kappa = 0.2 \pm 0.02$) for a 40% organosulfate, 60% 2-methyltetrol composition ($\kappa$=0.23).

Figure 11e shows two observations of aerosol liquid water content compared to model predictions. In the model, aerosol LWC was represented as the sum of water due to inorganic species ($W_i$, referred to as inorganic water) and water due to organic species ($W_o$, referred to as organic water). The LLPS and Base simulations resulted in the same predictions of aerosol water as

only inorganic species were considered in calculating LWC. The difference between the Base simulation and observed LWC indicate a potential role for water associated with organic species. The contribution of LWC due to organic species has been estimated as 35% during SOAS with higher contirubutions (50%) at night (Guo et al., 2015). Both organic and inorganic water were predicted to be highest in concentration during the night or early morning as a result of the diurnal variation in RH.





Both simulations with uptake of water onto organic species ($W_o > 0$ and $\gamma \neq 1$) overpredicted LWC at night with the $W_o > 0$ simulation resulting in greater overprediction as a result of the feedback mentioned earlier. Figure 11f attributes the overprediction in organic water for the $\gamma \neq 1$ simulation to errors in the concentration of OA, hygroscopicity parameter for organic aerosol ($\kappa_{org}$), and $a_w$ (or RH) (equation 12). For simplicity in the attribution analysis, RH was converted to acitivity using a fixed particle diameter of 200 nm (Hu et al., 2016). $W_o$ was not directly measured, but estimated using measured properties. Figure 11f indicates overestimates in the concentration of OA at night resulted in overestimates in $W_o$. Underestimates in RH and $\kappa_{org}$ decreased the overestimate. Thus, predictions of aerosol water in the sensitivity simulations can be most improved by improving the concentration of OA in the model.

The concentration of organic water and contribution to total aerosol water is shown across the model domain in Figure 12. $W_o$ was generally predicted to peak in the same locations where OA (Figure 12c) was high. This trend was not true in locations where RH drove higher or lower water uptake than expected or OA was dominated by fresh POA with low OM/OC. For example, high RH over the Great Lakes and off the Northeast Coast resulted in high concentrations of organic water. High concentrations of OA from fires in Colorado did not translate to high aerosol water as a result of low RH and low OM/OC ratios leading to low $\kappa_{org}$. $\kappa_{org}$ was lower in urban areas as well (near 0.09) due to low OM/OC. Regionally, $\kappa_{org}$ ranged between 0.11 and 0.14. The contribution to aerosol water resulting from organic vs. inorganic species (Figure 12b) reflected the ratio of organic-to-sulfate concentrations as aerosol water is proportional to their concentrations.

### 3.2.5 Model relationship to WSOC

The Particle-into-Liquid Sampler (PiLS) instrument used to measure $WSOC_p$ adds an equivalent volume of water of $6 \times 10^6$ µgm$^{-3}$air which is significantly higher than the concentration of aerosol water observed during SOAS at CTR (less than 73 µgm$^{-3}$ (Nguyen et al., 2015b; Guo et al., 2015)). Figure 13a shows the fraction of particulate OA present in the aqueous (vs. insoluble) phase (equation 26). For the PiLS instrument during SOAS, compounds with $\gamma_i^\infty <$100,000 (solubilities as low as 0.1 g/L) were expected to be part of measured $WSOC_p$. Biogenic-VOC derived SOA was particularly soluble, except for potentially $MTNO_3$. Alkane and aromatic SOA had 1,000$<\gamma_i^\infty<$100,000 (0.1<S<10), and thus, were less soluble. Note that none of the species have very low solubilities, so all SOA species were expected to be at least partially water-soluble during extraction depending on ambient conditions. Using the PiLS estimate of fraction water-soluble OA of 90% (Washenfelder et al., 2015), the mole-weighted $\gamma_i^\infty$ for ambient OA was predicted (equation 26) to be 10,000,000, much higher than the coefficient predicted for any individual semivolatile constituent in the model.

The base simulation provided a good representation of $WSOC_p$ at night, but underestimated total OC at all hours of the day, particularly during the daytime. $\gamma \neq 1$ provided a better estimate of total OC, but overpredicted $WSOC_p$ at night if compounds with $\gamma^\infty < 1,000$ (S>10 g/L) were entirely considered WSOC. Recall that the *a priori* estimate of solubility for $MTNO_3$ was increased by a factor of 100 to reconcile modeled and observed LO-OOA and particulate organic nitrate for $\gamma \neq 1$. Even with the factor of 100 increase in Henry's law, $MTNO_3$ remained the least soluble biogenically-derived SOA species in the model. The large increase in OC for the nonideal simulation was a result of compounds with solubilities greater than 10 g/L or $\gamma^\infty < 1,000$ (Figure 13c) which were dominated by traditional biogenic SOA and its accretion products. The accretion product





from traditional semivolatile SOA is not well constrained in terms of its structure or volatility. In this work, as in the work of Carlton et al. (2010) and Pankow et al. (2015), the species was assumed nonvolatile with an OM/OC of 2 to 2.1. The solubility of low-$NO_x$ monoterpene-derived species remained above 10 g/L, even down to species with a saturation concentration of $1 \times 10^{-10}$ $\mu g m^{-3}$ using Henry's law coefficient values from Hodzic et al. (2014). If the accretion products (AOLGB) were

5 better represented by a less functionalized species and effectively insoluble, observation/model disagreement in $WSOC_p$ in $\gamma \neq 1$ would be reduced. Another way to reconcile observed and modeled $WSOC_p$ may be to take into account deviations from equilibrium during PiLS extraction, which were not considered here.

Figure 14 shows observed water soluble organic carbon compounds in the gas phase ($WSOC_g$, measured by Mist Chamber and Total Carbon Analzyer (Hennigan et al., 2009)) compared to (a) semivolatile SOA precursors and (b) semivolatile and

10 aqueous SOA precursors currently in CMAQ. The figure indicates that considering semivolatile SOA precursors as the only source of $WSOC_g$ in the model underestimated the daytime amount of $WSOC_g$, but that both observed $WSOC_g$ and modeled semivolatile SOA precursors were on the same order of magnitude. Thus, the semivolatile surrogates in the model represented a significant pool of soluble gases. When IEPOX, glyoxal, and methylglyoxal were included in the model estimate of $WSOC_g$, the daytime $WSOC_g$ was slightly overestimated. However, given the factor of 2.4 overestimate in IEPOX+ISOPOOH in the

15 model compared to observations (supporting information), the speciation of $WSOC_g$ differed in the model and observations. Figure 14 indicates that during the daytime, either additional water-soluble SOA precursors need to be implemented in the model or the model is correct and a significant portion of ambient $WSOC_g$ does not lead to SOA. Indeed, observed $WSOC_g$ may have large contributions from compounds such as formic acid that are not considered significant SOA constituents (Liu et al., 2012).

## 4   Conclusions

Current chemical transport models consider the dominant pathways to SOA to be dry processes governed by condensation of low-volatility organic compounds in the absence of water. In addition, models generally do not consider uptake of water by organic species. In this work, the CMAQ model was updated to consider aerosol water interactions with semivolatile SOA species and uptake of water onto OA with a focus on simulating conditions during the Southern Oxidant and Aerosol Study

of 2013. A method was developed to take into account deviations from ideality using an activity coefficient calculated based on the species Henry's law coefficient, pure species saturation concentration ($C_{0,i}^*$), and mole fraction water in the particle that resulted in a normalized mean bias of -4%, -10%, and -2% for IMPROVE, CSN, and SEARCH OC. Monoterpene nitrates were predicted to be the least soluble semivolatile in the model, consistent with SOA yields from $\beta$-pinene+$NO_3$ being comparable under dry and humid conditions (Boyd et al., 2015). However, most biogenic hydrocarbon-derived semivolatile SOA was

highly soluble and predicted to be measured as WSOC. Thus, even aerosol formed through dry processes in models may be classified as WSOC as measured by instruments such as the PiLS.

Based on current observations, aerosol water cannot be added to the partitioning medium for semivolatile organic compounds without simultaneously accounting for deviations in ideality. Otherwise, aerosol liquid water and aerosol carbon is overesti-



mated at night. This finding is consistent with the work by Pun (2008) who found that aerosol water concentrations would more than double if ideality was assumed. Hodas et al. (2015) also found that organic-inorganic water-uptake experiments could not be modeled assuming ideal, well-mixed liquids, and assuming ideality overpredicted $\alpha$-pinene SOA concentrations by 100-200% in the work of Zuend and Seinfeld (2012).

All simulations in this work, including the more aggressive ones assuming ideality, could not reproduce daytime observed OA in the southeast US (at SEARCH sites) solely by adding water to the partitioning medium. Including water resulted in increased model error but could reduce the bias in OC. Additional pathways (new precursors and/or new pathways) to OA, particularly during the daytime, are still needed in models.

The updates described here are in three stages of model-readiness:

1. Properties of semivolatile OA constituents can immediately be updated in models to be consistent with their assumed volatility and parent hydrocarbon. Base model performance was good in terms of isoprene OA and total OC compared to routine networks. Property updates in this work are scheduled for public release as part of CMAQv5.2.

2. Prediction of organic water is more uncertain, but OM/OC is a useful proxy and can be used to parameterize water uptake onto organic species via equation 11 and $\kappa$-Köhler theory.

3. The effects of water on semivolatile OA partitioning requires additional research as deviations from ideality are important. $\gamma_i^\infty$ or $C_{H,i}^*$ are recommended as useful parameters for characterizing solubility. Models such as the Aerosol Inorganic-Organic Mixtures Functional groups Activity Coefficients (AIOMFAC) model (Zuend et al., 2008) and UMan-SysProp (Topping et al., 2016) offer opportunities to perform detailed calculations.

In addition, these areas of model improvement are suggested for future work:

1. A treatment of semivolatile primary OA is needed to reproduce observed surrogates for POA. Factor of 2 overestimates in POA were predicted to compensate for underestimates in SOA on the order of 40% in IMPROVE and CSN networks.

2. Improvements to sulfate and gas-phase isoprene chemistry will lead to an improved isoprene OA representation in models as isoprene OA is correlated with sulfate, but precursors to IEPOX-derived SOA were overestimated at CTR during SOAS. Predictions of isoprene SOA could be further improved by considering the volatility of IEPOX-derived
species (such as 2-methyltetrols and $C_5$-alkene triols) as well as formation of additional species (Riedel et al., 2016).

3. Model predicted aerosol LWC that includes water associated with organic compounds can most be improved by improving the concentration of OA which may require a number of updates in different areas.

4. New precursors to SOA are likely needed, especially during the day when OA is underestimated and gas-phase semivolatile model species are less plentiful. Additional precursors for the isoprene system may include multifunctional hydroperox-
ides (Riva et al., 2016).





## 5 Data availability

CMAQv5.1 is publicly available via github (https://github.com/CMASCenter/CMAQ/) and the Community Modeling and Analysis System (CMAS) Center (https://www.cmascenter.org/cmaq/). Property updates are scheduled for public release as part of CMAQv5.2. SOAS field data is available from http://esrl.noaa.gov/csd/groups/csd7/measurements/2013senex/.

5  *Acknowledgements.* We thank CSC for emission processing and Shaojie Song for GEOS-Chem simulations. We thank Jesse Bash, Donna Schwede, and Matt Woody for useful discussion and Kirk Baker for developing the SOAS modeling platform. We thank William H. Brune, David O. Miller, Philip A. Feiner, and Li Zhang for providing OH data. We thank Paul Wennberg and Tran Nguyen for providing CTR CIMS data for IEPOX+ISOPOOH. We thank Rohit Mathur and Golam Sarwar for providing comments on the manuscript. The U.S. EPA through its Office of Research and Development supported the research described here. It has been subjected to Agency administrative review and

10  approved for publication, but may not necessarily reflect official Agency policy. The Ng group was supported by NSF grant 1455588 and EPA grant RD-83540301. RW was supportedby NSF grant 1242258. WWH and JLJ were supported by EPRI 10004734, NSF AGS-1360834 and EPA STAR 83587701-0. AGC was supported by EPA STAR grant R83512 and NSF-AGS 1242155. The Surratt group was supported by EPRI as well from EPA grant R835404 and NSF Grant CHE-1404644. A.H. Goldstein the UC Berkeley team acknowledge support from EPA STAR Grant R835407 and NSF Grant AGS-1250569. G.I.VW is supported by the NSF Graduate Research Fellowship (#DGE 1106400).



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



**Table 1.** SOA and SVOC species in CMAQ v5.1-aero6i (Carlton et al., 2010; Pye and Pouliot, 2012; Pye et al., 2013, 2015). CMAQ model species names are generally preceeded by the letter A to indicate aerosol. Semivolatile surrogates have a corresponding gas-phase species whose name is preceeded by the letters SV.

| Species | Production pathway description | Partitioning medium |
|---------|-------------------------------|---------------------|
| ALK1 | alkane + OH SOA/SVOC | Dry organic aerosol |
| ALK2 | alkane + OH SOA/SVOC | Dry organic aerosol |
| BNZ1 | benzene + OH high-$NO_x$ SOA/SVOC | Dry organic aerosol |
| BNZ2 | benzene + OH high-$NO_x$ SOA/SVOC | Dry organic aerosol |
| BNZ3 | benzene + OH low-$NO_x$ SOA | Dry organic aerosol |
| DIM | IEPOX-derived dimers | Aqueous aerosol |
| GLY | glyoxal + methylglyoxal SOA | Aqueous aerosol |
| IEOS | IEPOX-derived organosulfate | Aqueous aerosol |
| IETET | 2-methyltetrols | Aqueous aerosol |
| IMGA | 2-methylglyceric acid | Aqueous aerosol |
| IMOS | MPAN-derived organosulfate | Aqueous aerosol |
| ISO1 | isoprene+OH SOA/SVOC | Dry organic aerosol |
| ISO2 | isoprene+OH SOA/SVOC | Dry organic aerosol |
| ISO3 | acid-catalyzed isoprene SOA[a] | Dry organic aerosol |
| ISOPNN | isoprene dinitrate | Dry organic aerosol |
| MTHYD | organic nitrate hydrolysis product | Aqueous aerosol (from dry organic aerosol parent) |
| MTNO3 | monoterpene nitrate | Dry organic aerosol |
| OLGA | oligomers from anthropogenic SOA/SVOCs | Dry organic aerosol |
| OLGB | oligomers from biogenic SOA/SVOCs | Dry organic aerosol |
| ORGC | glyoxal+methylglyoxal SOA | Cloud droplets |
| PAH1 | naphthalene + OH high-$NO_x$ SOA/SVOC | Dry organic aerosol |
| PAH2 | naphthalene + OH high-$NO_x$ SOA/SVOC | Dry organic aerosol |
| PAH3 | naphthalene + OH low-$NO_x$ SOA | Dry organic aerosol |
| SQT | sesquiterpene + OH, $O_3$, $NO_3$, $O_3P$ SOA/SVOC | Dry organic aerosol |
| TOL1 | toluene + OH high-$NO_x$ SOA/SVOC | Dry organic aerosol |
| TOL2 | toluene + OH high-$NO_x$ SOA/SVOC | Dry organic aerosol |
| TOL3 | toluene + OH low-$NO_x$ SOA | Dry organic aerosol |
| TRP1 | monoterpene + OH, $O_3$, $O_3P$ SOA/SVOC | Dry organic aerosol |
| TRP2 | monoterpene + OH, $O_3$, $O_3P$ SOA/SVOC | Dry organic aerosol |
| XYL1 | xylene + OH high-$NO_x$ SOA/SVOC | Dry organic aerosol |
| XYL2 | xylene + OH high-$NO_x$ SOA/SVOC | Dry organic aerosol |
| XYL3 | xylene + OH low-$NO_x$ SOA | Dry organic aerosol |

[a]AISO3 contains the sum of 2-methyltetrols and IEPOX-derived organosulfates in CMAQv5.1-aero6. It is not used in aero6i as those species are represented individually. Prior to v5.1, AISO3 was determined as an enhancement over AISO1+AISO2 based on [$H^+$] (Carlton et al., 2010).



**Table 2.** *A priori* SOA and SVOC properties. All temperature dependent parameters given at 298 K.

| Species | $C_0^*$ µgm⁻³ | α gg⁻¹ | OM/OC gg⁻¹ | $\widetilde{M}$ gmol⁻¹ | $n_C$ | H Matm⁻¹ | $D_g$ cm²s⁻¹ | $V_{LeBas}$ cm³mol⁻¹ | κ | ρ kgm⁻³ | $\gamma^{\infty,f}$ | S gL⁻¹ | $C_H^{*\,f}$ µgm⁻³ |
|---|---|---|---|---|---|---|---|---|---|---|---|---|---|
| ALK1 | 0.1472[a] | 0.0334 | 1.56 | 225 | 12 | $6.2 \times 10^8$ | 0.0514 | 280.5 | 0.07 | 1400 | 5600 | 2 | $8.3 \times 10^2$ |
| ALK2 | 51.8775[a] | 0.2164 | 1.42 | 205.1 | 12 | $4.5 \times 10^6$ | 0.0546 | 275.6 | 0.06 | 1400 | 2000 | 6 | $1.0 \times 10^5$ |
| BNZ1 | 0.302[b] | 0.0720 | 2.68[c] | 161 | 5 | $2.1 \times 10^8$ | 0.0642 | 134.1 | 0.19 | 1400 | 5800 | 2 | $1.7 \times 10^3$ |
| BNZ2 | 111.11[b] | 0.8880 | 2.23[c] | 134 | 5 | $2.0 \times 10^6$ | 0.0726 | 127.5 | 0.15 | 1400 | 1400 | 5 | $1.5 \times 10^5$ |
| BNZ3 | NA | 0.370 | 3.00[c] | 180 | 5 | NA | NA | NA | 0.23 | 1400 | NA | <1 | NA |
| DIM | NA | NA | 2.07[d] | 248.2 | 10 | NA | NA | NA | 0.13 | 1400 | NA | NA | NA |
| GLY | NA | NA | 2.13[e] | 66.4 | 3 | NA | NA | NA | 0.13 | 1400 | NA | NA | NA |
| IEOS | NA | NA | 3.60[d] | 216.2 | 5 | NA | NA | NA | 0.30 | 1400 | NA | NA | NA |
| IETET | NA | NA | 2.27[d] | 136.2 | 5 | NA | NA | NA | 0.15 | 1400 | NA | NA | NA |
| IMGA | NA | NA | 2.50[d] | 120.1 | 4 | NA | NA | NA | 0.18 | 1400 | NA | NA | NA |
| IMOS | NA | NA | 4.17[d] | 200.2 | 4 | NA | NA | NA | 0.36 | 1400 | NA | NA | NA |
| ISO1 | 116.01[b] | 0.2320 | 2.20[c] | 132 | 5 | $4.3 \times 10^7$ | 0.0733 | 126.3 | 0.14 | 1400 | 60 | 120 | $6.9 \times 10^3$ |
| ISO2 | 0.617[b] | 0.0288 | 2.23[c] | 133 | 5 | $3.7 \times 10^9$ | 0.0729 | 123.8 | 0.15 | 1400 | 130 | 56 | $8.2 \times 10^1$ |
| ISO3 | NA | NA | 2.80[d] | 168.2 | 5 | NA | NA | NA | 0.21 | 1400 | NA | NA | NA |
| ISOPNN | 8.9[e] | NA | 3.80[e] | 226 | 5 | $4.5 \times 10^{8\,e}$ | 0.0457[e] | 206.8[e] | 0.32 | 1400 | 130 | 98 | $1.1 \times 10^3$ |
| MTHYD | NA | NA | 1.54[e] | 185 | 10 | NA | NA | NA | 0.07 | 1400 | NA | NA | NA |
| MTNO3 | 12[e] | NA | 1.90[e] | 231 | 10 | $1.5 \times 10^{6\,e,f}$ | 0.0453[e] | 251.2[e] | 0.11 | 1400 | 29000 | 0.4 | $3.5 \times 10^5$ |
| OLGA | NA | NA | 2.50[c] | 206 | 7 | NA | NA | NA | 0.18 | 1400 | NA | <1 | NA |
| OLGB | NA | NA | 2.10[c] | 248 | 10 | NA | NA | NA | 0.13 | 1400 | NA | >10 | NA |
| ORGC | NA | NA | 2.00[b] | 177 | 7 | NA | NA | NA | 0.12 | 1400 | NA | NA | NA |
| PAH1 | 1.6598[a] | 0.2100 | 1.63 | 195.6 | 10 | $5.1 \times 10^7$ | 0.0564 | 235.7 | 0.08 | 1480 | 5300 | 2 | $8.8 \times 10^3$ |
| PAH2 | 264.6675[a] | 1.0700 | 1.49 | 178.7 | 10 | $7.2 \times 10^5$ | 0.0599 | 231.5 | 0.06 | 1480 | 2100 | 5 | $5.7 \times 10^5$ |
| PAH3 | NA[a] | 0.7300 | 1.77 | 212.2 | 10 | NA | NA | NA | 0.09 | 1550 | NA | <1 | NA |
| SQT | 24.984[b] | 1.5370 | 1.52[c] | 273 | 15 | $6.2 \times 10^8$ | 0.0451 | 346.5 | 0.07 | 1400 | 40 | 380 | $1.0 \times 10^3$ |
| TOL1 | 2.326[b] | 0.0580 | 2.26[c] | 163 | 6 | $4.2 \times 10^7$ | 0.0637 | 153.7 | 0.15 | 1240 | 3800 | 2 | $8.9 \times 10^3$ |
| TOL2 | 21.277[b] | 0.1130 | 1.82[c] | 175 | 8 | $7.3 \times 10^6$ | 0.0607 | 194.1 | 0.10 | 1240 | 2600 | 4 | $5.5 \times 10^4$ |
| TOL3 | NA | 0.300 | 2.70[c] | 194 | 6 | NA | NA | NA | 0.20 | 1450 | NA | <1 | NA |
| TRP1 | 14.792[b] | 0.1393 | 1.84[c] | 177 | 8 | $9.9 \times 10^8$ | 0.0603 | 194.9 | 0.10 | 1400 | 27 | 360 | $4.0 \times 10^2$ |
| TRP2 | 133.7297[b] | 0.4542 | 1.83[c] | 198 | 9 | $1.4 \times 10^8$ | 0.0559 | 218.8 | 0.10 | 1400 | 25 | 450 | $3.3 \times 10^3$ |
| XYL1 | 1.314[b] | 0.0310 | 2.42[c] | 174 | 6 | $6.2 \times 10^7$ | 0.061 | 154.6 | 0.17 | 1480 | 4900 | 2 | $6.4 \times 10^3$ |
| XYL2 | 34.483[b] | 0.0900 | 1.93[c] | 185 | 8 | $4.0 \times 10^6$ | 0.0585 | 194.6 | 0.11 | 1480 | 3100 | 3 | $1.1 \times 10^5$ |
| XYL3 | NA | 0.360 | 2.30[c] | 218 | 8 | NA | NA | NA | 0.15 | 1330 | NA | <1 | NA |

[a]Pye and Pouliot (2012). [b]Carlton et al. (2010). [c]Pankow et al. (2015). [d]Pye et al. (2013). [e]Pye et al. (2015). [f]A factor of 100 increase in MTNO3 Henry's law coefficient, factor of 10 decrease in $\gamma^\infty$, and factor of 10 decrease in $C_{x_w=1}^*$ produced better model results in the $\gamma \neq 1$ simulation. See supporting information for *a posteriori* $\gamma \neq 1$ simulation parameters. NA indicates not applicable (nonvolatile species).





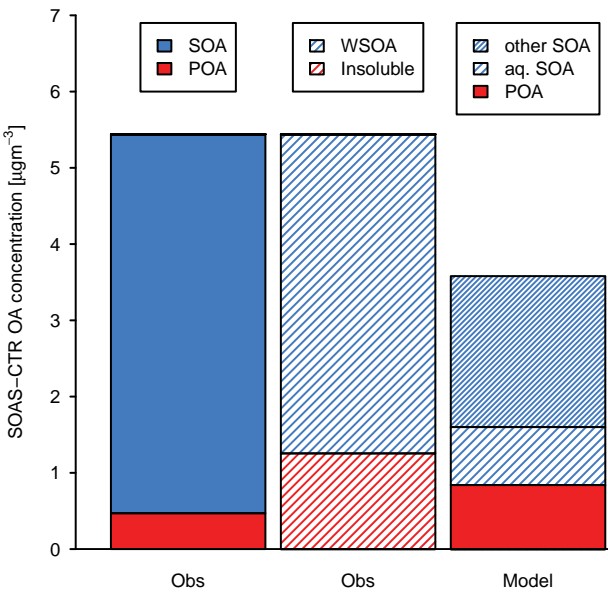

**Figure 1.** Contribution of POA (observed biomass burning OA, BBOA, Xu et al. (2015a)), SOA, water-soluble OA (estimated as 2.1 × WSOC from the PiLS, Sullivan et al. (2004)), and aqueous SOA (model only) to total OA during June 2013 observed at CTR during SOAS and modeled by CMAQ.



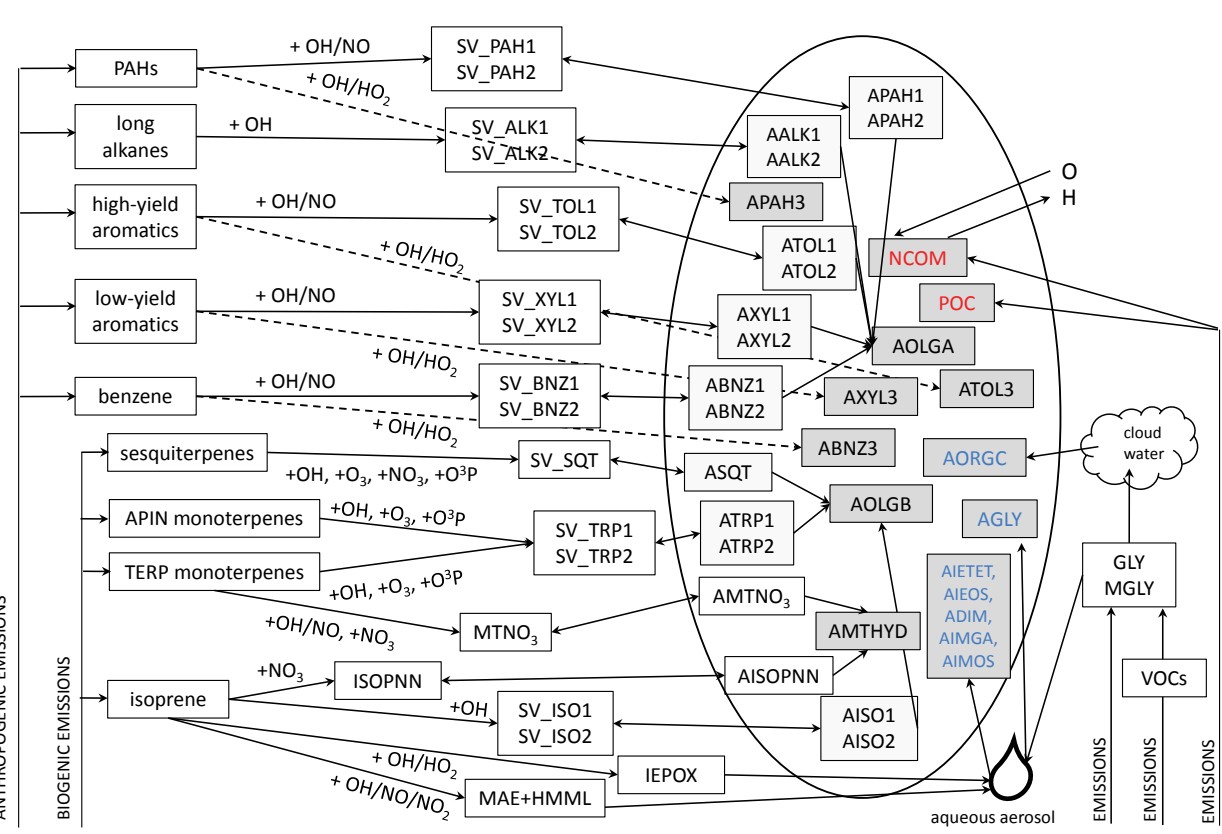

**Figure 2.** Schematic of SOA treatment in current CMAQ-aero6i. Species are described in Table 1. Species in grey boxes are nonvolatile. Species with names in red make up POA. Species with names in blue form in the model as a direct result of interactions with water.




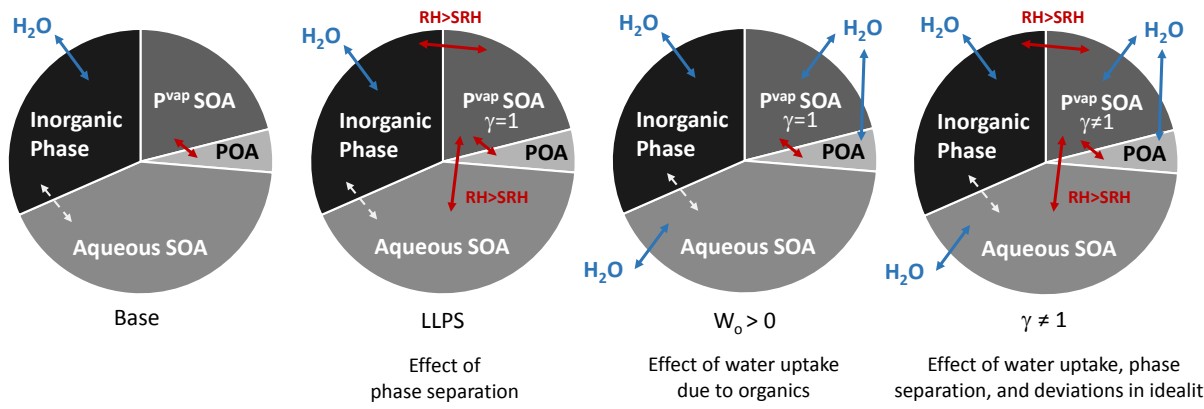

**Figure 3.** Interactions of the inorganic phase (e.g. sulfate, nitrate, ammonium, aerosol water), aqueous SOA, vapor-pressure driven SOA, and POA in the base and sensitivity simulations. Blue arrows depict water partitioning/uptake. Red arrows indicate semivolatile partitioning interactions. The white dashed arrows indicate aqueous SOA interaction with the inorganic phase (via liquid water, acidity, etc).



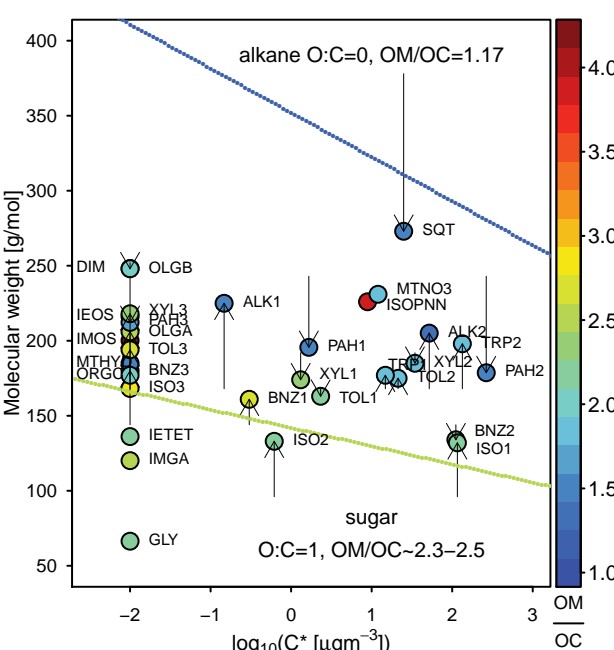

**Figure 4.** The volatility, molecular weight, and OM/OC of SOA species in CMAQ. Nonvolatile species are arbitrarily plotted at a saturation concentration of $0.01\ \mu g m^{-3}$. The arrows start at the old molecular weights assumed in CMAQv5.1. The arrows end at the new molecular weights in Table 2. Lines indicate the properties of alkanes and sugars.



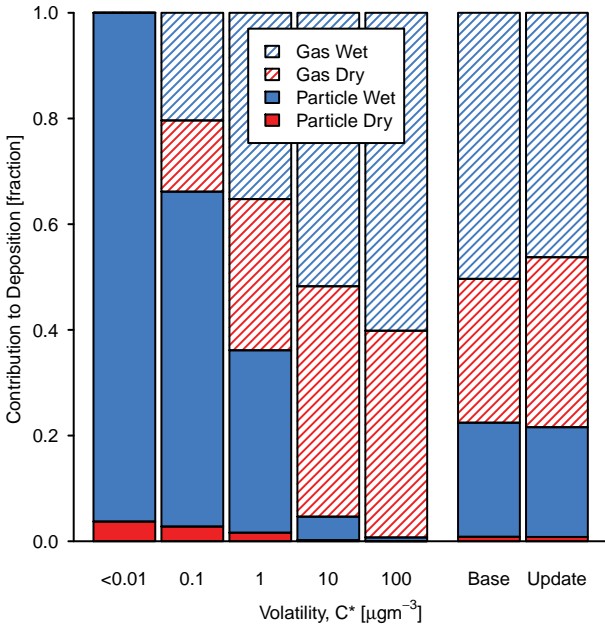

**Figure 5.** Contribution of wet (blue) and dry (red) deposition of gas (striped) and particle (solid) SVOCs binned by volatility and overall compared to the base simulation (CMAQ v5.1). Nonvolatile species are indicated by $C^*$<0.01 μgm$^{-3}$. POA is not included.

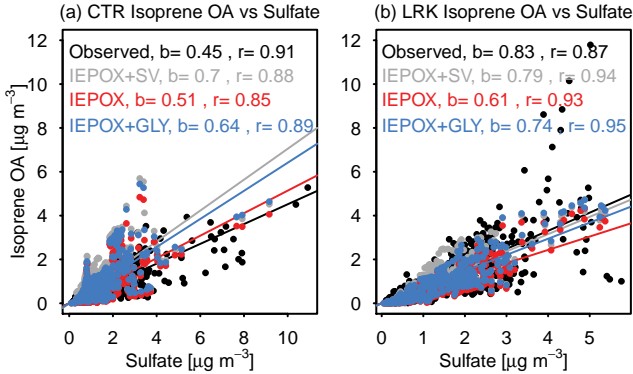

**Figure 6.** Isoprene OA vs sulfate at (a) CTR and (b) LRK and the slope (b, forced through 0) and correlation coefficient (r) for each data set. Model representations of isoprene OA include SOA from IEPOX uptake and semivolatile isoprene+OH SOA (IEPOX+SV), SOA from IEPOX uptake (IEPOX), and SOA from IEPOX and glyoxal uptake (IEPOX+GLY).





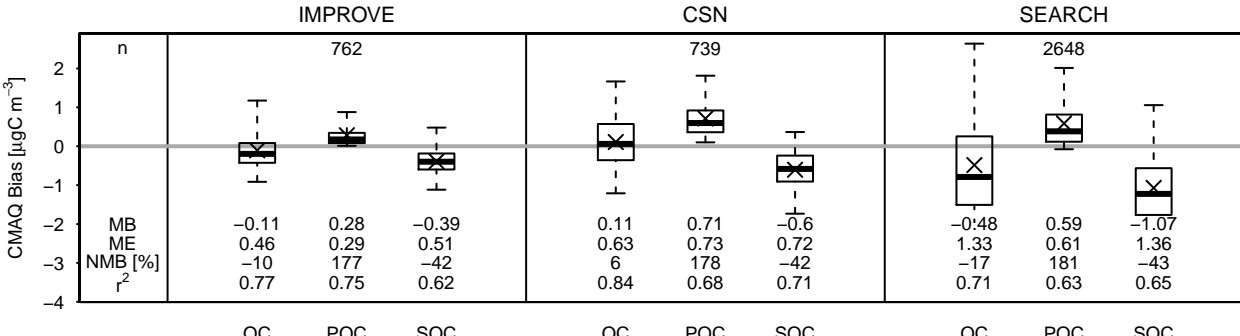

**Figure 7.** Aerosol OC, POC, and SOC predicted by the base simulation compared to CSN, IMPROVE, and SEARCH (JST, BHM, CTR, and YRK) observations. Mean bias (MB) and mean absolute gross error (ME) are in $\mu g C m^{-3}$. X symbols indicate mean bias. Boxplots indicate 5th, 25th, median, 75th, and 95th percentile. $r^2$ based on a zero intercept.

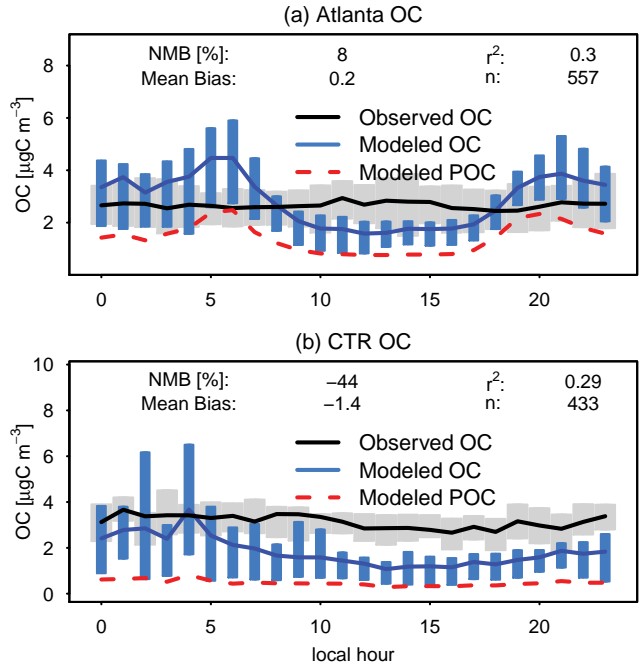

**Figure 8.** OC as a function of hour of the day for a SEARCH urban (Atlanta, JST) and rural (CTR) site during June 2013. Bars/shading indicate 25th to 75th percentiles. Lines indicate means. Red dashed lines indicate model-predicted POC.



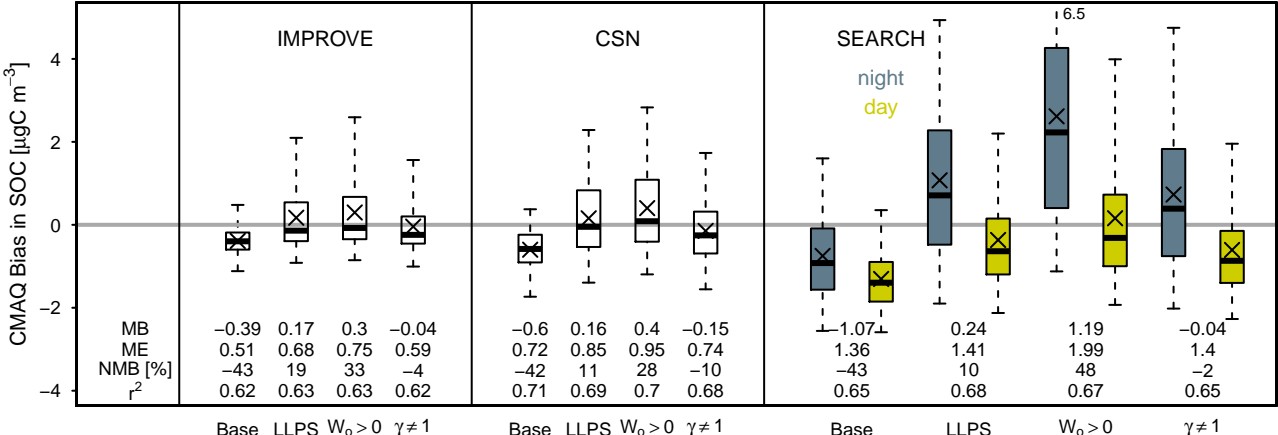

**Figure 9.** Bias (model-observation) in SOC for IMPROVE, CSN, and SEARCH networks. SEARCH data is divided into daytime (6:00 am to 7:59 pm local time) and nighttime observations. SOC is calculated using OC/EC ratios and estimating evaporation of semivolatiles as described in section 2.9. X symbols indicate mean bias. Boxplots indicate 5th, 25th, median, 75th, and 95th percentile. $r^2$ based on a zero intercept.

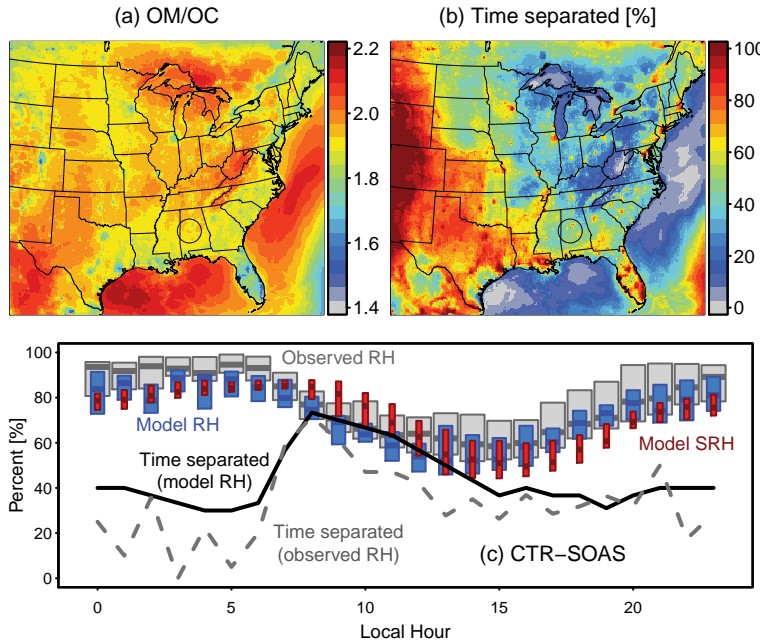

**Figure 10.** June 2013 mean predicted (a) OM/OC, (b) percentage of time spent separated into organic-rich and inorganic-rich phases, and (c) conditions at CTR-SOAS for the LLPS simulation. Separation occurs when RH<SRH. Observed RH at SOAS is from the SEARCH network. Panel (c) includes a prediction of time separated using model predicted RH (solid) and observed RH (dashed).



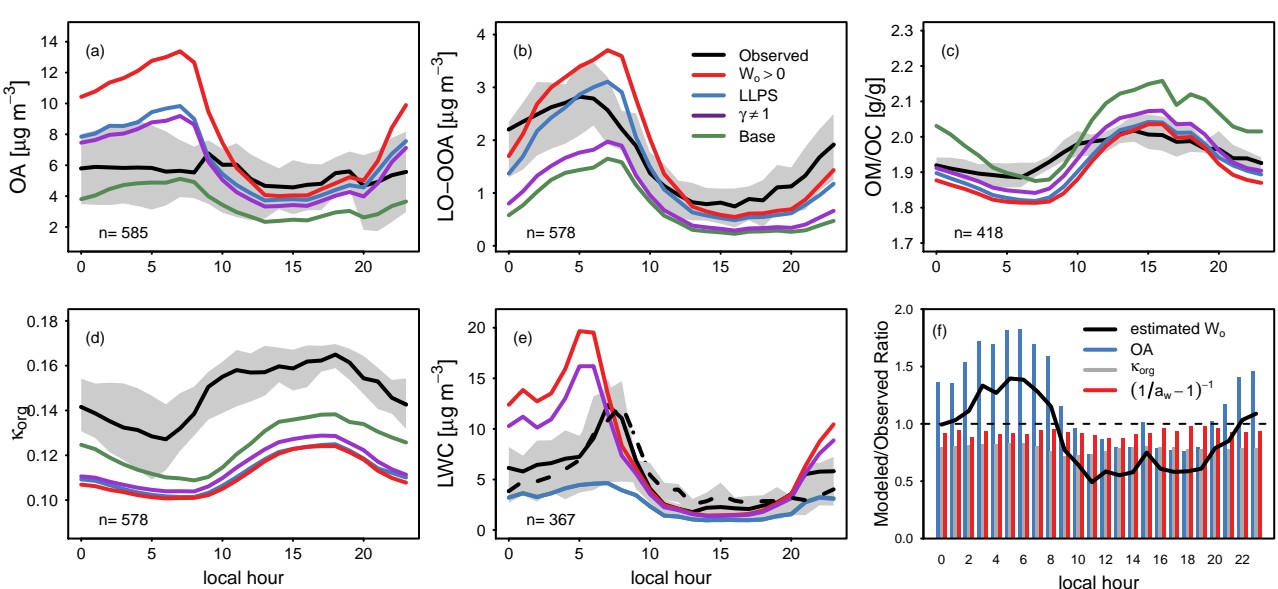

**Figure 11.** Observed and predicted concentration of (a) organic aerosol (Xu et al., 2015a), (b) AMS LO-OOA factor (Xu et al., 2015a) and model predicted organic nitrate-derived SOA, (c) OM/OC (Hu et al., 2015), (d) $\kappa_{org}$, (e) aerosol liquid water measured by nephelometer and the Georgia Tech group (solid black) (Guo et al., 2015) and measured by particle growth and the Rutgers/NC State group (dashed black) (Nguyen et al., 2014b), and (f) ratio of predicted to observed quantities influencing organic water ($W_o$) at CTR (for the $\gamma \neq 1$ simulation only). Observed $\kappa_{org}$ is determined by applying a $\kappa_{org,i}$ value of 0.31, 0.20, 0.16, and 0.08 to observed BBOA, ISOPOA, MO-OOA, and LO-OOA respectively (Cerully et al., 2015)). Grey shading represents the interquartile range of the observed data (mean in black). Colors represent different simulations in a-e and different quantities in f.




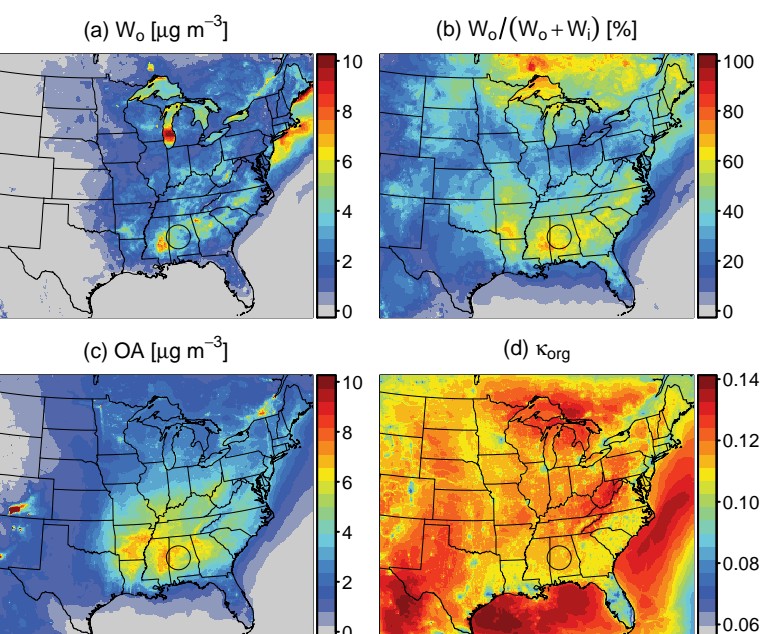

**Figure 12.** June 2013 mean predicted (a) aerosol water due to organic species, (b) contribution of organic water to total aerosol water, (c) total organic aerosol, and (d) hygroscopicity parameter for the $\gamma \neq 1$ simulation.



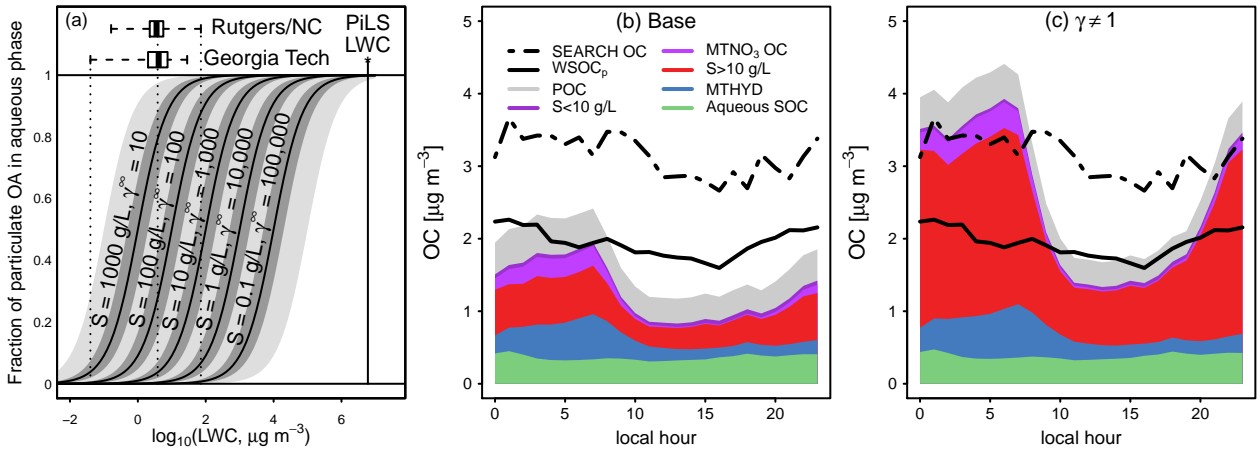

**Figure 13.** Fraction of OA present in aqueous phase (a) as a function of activity coefficient at infinite dilution and observed OC and $WSOC_p$ at CTR compared to model predictions (b-c). Panel (a) boxplots indicate observed LWC from Rutgers/NC State (Nguyen et al., 2014b) and Georgia Tech (Guo et al., 2015) during SOAS. For predictions (a), WIOA is 1 $\mu$gm$^{-3}$ and the species molecular weight is set to 180 g/mol. Predictions in dark grey shading span a factor of 2 in WIOA (0.5 to 2 $\mu$gm$^{-3}$). Predictions in light grey shading (a) indicate a factor of 10 in WIOA (0.1 to 10 $\mu$gm$^{-3}$). Panel (b) corresponds to model predictions in the base simulation while panel (c) corresponds to predictions in the $\gamma \neq 1$ simulation. Model predictions of OC are stacked and divided into POC, compounds with $\gamma_i^\infty > 1,000$ (solubilities less than 10 g/L, Table 2), monoterpene nitrate OC, compounds with $\gamma_i^\infty < 1,000$ (solubilities greater than 10 g/L), the organic nitrate hydrolysis product, and aqueous SOC.





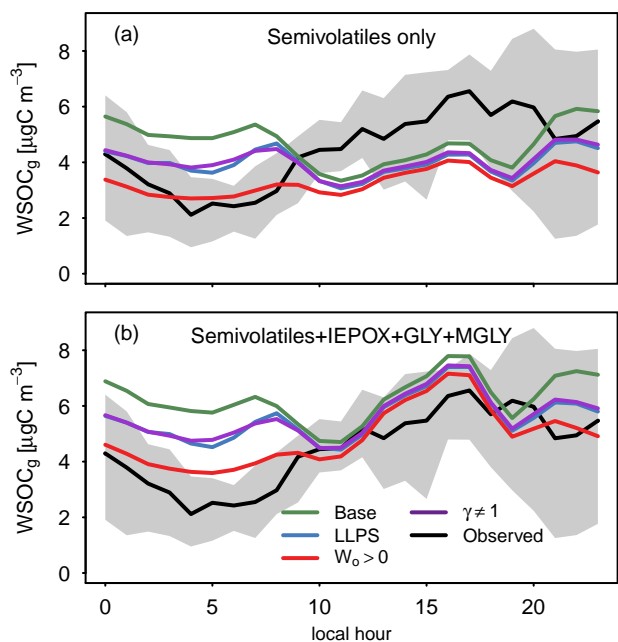

**Figure 14.** Observed $WSOC_g$ (Hennigan et al., 2009; Xu et al., in preparation) and model SOA precursors considering only semivolatile surrogates (a) and semivolatile and aqueous surrogates (b). Grey shading represents the interquartile range of the observed data (mean in black). Colors represent different simulations.