# Peer review of "On the implications of aerosol liquid water and phase separation for organic aerosol mass"

_Atmospheric Chemistry and Physics, 2016_

## Referee Comment (RC1) · Anonymous Referee #1 · 15 Oct 2016

This study investigated aerosol water interactions with semi-volatile SOA species and uptake of water onto OA with a focus on simulating conditions during the SOAS campaign. They have developed a method to take into account effects of non-ideality and phase separation on SOA formation and partitioning. They found that inclusion of water in an organic phase led to increased SOA concentrations, particularly at night. Aerosol properties, such as the OM/OC and hygroscopicity parameter were captured well, but additional improvements in daytime organic aerosol are needed to close the model-measurement gap. I found this modeling study is conducted very well making use of comprehensive measurements conducted in the SOAS campaign. The authors made great efforts in implementing state-of-the-art knowledge of aerosol water and phase separation into the regional model. I appreciate this study very much and recommend for publication in ACP after the following comments are addressed.

[Figure]

Specific comments:

- P4, L10: Have you actually implemented eq(1) into the model? Terpene and toluene SOA have higher viscosity and lower bulk diffusivity compared to isoprene SOA (e.g., Renbaum-Wolff et al., 2013; Song et al., ACP, 2016). Were SOA particles in SOAS totally dominated by isoprene SOA and only little influence by terpene or anthropogenic emissions? Please clarify and justify.

- P4, L21: "SOA from cloud processing is relatively minor in terms of average SOA concentrations.": Please justify this sentence by adding some explanations or reference(s). Have you investigated this by modeling or is this implied by measurements? Was aqueous phase processing (particle-phase chemistry in deliquesced particles) also minor?

- P5, L13: Just to draw your attention, the recent study (Li et al., 16, 3327, ACP, 2016) has extended eq(2) by including number of N and S per molecule to estimate volatility.

- P5, eq(2): This should be $C_0$, but not $C^*$. As you discuss in this manuscript, $C^*$ (effective saturation mass concentration) includes the effect of non-ideality, but I believe eq(2) can be only used to estimate pure compounds saturation mass concentration.

- Figure 4: I wonder why $C^*$ of glyoxal is 1e02 ug m-2. I believe $C_0$ of glyoxal is much higher based on eq(2). Is this because glyoxal is very water soluble so that $C^*$ is lowered?

- Figure 4: It seems that high molar mass and low volatility compounds (i.e., particle-phase products such as dimers, peroxyhemiacetals, oligomers, etc. that would locate close to the alkane line) were not considered in CMAQ. Do you have reason? Were such compounds not detected by measurements in SOAS?

- P13, L12, P14, L2: All IEPOX-derived species were assumed to be non-volatile and I suppose that the model also assumed that such product formation is irreversible. Have you evaluated this assumption by sensitivity simulations?Is it possible that such product

formation is actually reversible and might evaporate under certain conditions?

- P15, Figure 7: Model reproduced OC well, while overestimating POC and underestimating SOC. Is it possible actually this may suggest that measurements might have overestimated SOC? I suppose that SOC was assumed to be equivalent to OOA based on AMS-PMF analysis (correct?). What are uncertainties of the measurements? I wonder this, as AMS measures chemical composition and estimate secondary formation processes by post numerical analysis, while the model simulates secondary processes directly.

- P21, L8: Could gas-phase ELVOC formation or particle-phase chemistry help to close the measurement-model gap?

---

## Author Comment (AC1) · 26 Oct 2016

"On the implications of aerosol liquid water and phase separation for organic aerosol mass"
by Havala O. T. Pye et al.

Response to reviewer #1
We thank reviewer #1 for their comments and overall recommendation for publication. Our responses
are in blue with new text added to the manuscript underlined.

This study investigated aerosol water interactions with semi-volatile SOA species and
uptake of water onto OA with a focus on simulating conditions during the SOAS campaign.
They have developed a method to take into account effects of non-ideality and
phase separation on SOA formation and partitioning. They found that inclusion of water
in an organic phase led to increased SOA concentrations, particularly at night. Aerosol
properties, such as the OM/OC and hygroscopicity parameter were captured well, but
additional improvements in daytime organic aerosol are needed to close the model measurement
gap. I found this modeling study is conducted very well making use of
comprehensive measurements conducted in the SOAS campaign. The authors made
great efforts in implementing state-of-the-art knowledge of aerosol water and phase
separation into the regional model. I appreciate this study very much and recommend
for publication in ACP after the following comments are addressed.

- P4, L10: Have you actually implemented eq(1) into the model? Terpene and toluene
SOA have higher viscosity and lower bulk diffusivity compared to isoprene SOA (e.g.,
Renbaum-Wolff et al., 2013; Song et al., ACP, 2016). Were SOA particles in SOAS
totally dominated by isoprene SOA and only little influence by terpene or anthropogenic
emissions? Please clarify and justify.

Equation (1) for the diffusivity of IEPOX in aerosol was implemented in the model. This diffusivity is used
in the calculation of the uptake coefficient for IEPOX following Pye et al. (2013) and allows for a surface-
based process or bulk reaction depending on the timescale for diffusion relative to reaction (captured by
the parameter q):

$$\gamma = \left( \frac{1}{\alpha} + \frac{v}{4HRT\sqrt{D_a k_{particle}}} \frac{1}{f(q)} \right)^{-1}$$

$$f(q) = \coth(q) - 1/q$$

$$q = r_p \sqrt{\frac{k_{particle}}{D_a}}$$

As the reviewer points out, the diffusivity of species in the aerosol phase is influenced by both
composition and RH and we have only captured the RH dependence. SOAS was not dominated by
isoprene SOA. Xu et al. 2015 and Hu et al. 2015 both report that low NO isoprene SOA (IEPOX-SOA)
accounted for approximately 18% of total OA. Monoterpenes were also major contributors (Xu et al.,
2015). Urban POA and SOA is estimated to contribute ~28% of the OA, which is consistent with the fossil
carbon fraction (Kim et al., 2015), and there were also smaller contributions from biomass burning and
other sources. The most relevant aerosol diffusivity for IEPOX uptake would be that of IEPOX in a

multicomponent aerosol particle, thus using the value for monoterpene SOA is not necessarily a better approach than what we have used at this time. We have clarified that the diffusivity is used for IEPOX.

> The diffusivity of  IEPOX in the particle (Da, cm2s−1 ) was predicted by fitting a line through the data in the work of Song et al. (2015) resulting in

- P4, L21: "SOA from cloud processing is relatively minor in terms of average SOA concentrations.": Please justify this sentence by adding some explanations or reference(s). Have you investigated this by modeling or is this implied by measurements? Was aqueous phase processing (particle-phase chemistry in deliquesced particles) also minor?

Aqueous phase processing in deliquesced particles was significant as it is the major route to IEPOX SOA in the model. Extensive vertical profiling from aircraft during SENEX and SEAC4RS concluded that SOA formation in clouds was small and not statistically significant (Wagner et al., 2015). Cloud processing is also predicted to be relatively minor in CMAQ (although additional work is needed on this topic). SOA from cloud processing accounted for less than 3% of the organic aerosol concentration in our base simulation:

[Figure]

**Percent contribution of AORGC to total OM**

> SOA from cloud processing is predicted to result in less than 3% of total organic aerosol in CMAQ .

- P5, L13: Just to draw your attention, the recent study (Li et al., 16, 3327, ACP, 2016) has extended eq(2) by including number of N and S per molecule to estimate volatility.

Thank you for the reference. At this time, we do not use equation 2 for sulfate or nitrate containing organic compounds.

- P5, eq(2): This should be C0, but not C*. As you discuss in this manuscript, C* (effective saturation mass concentration) includes the effect of non-ideality, but I believe eq(2) can be only used to estimate pure compounds saturation mass concentration.

Correct. We have revised the equation (and text) to reflect it is for pure compounds.

> Donahue et al. (2011) developed a relationship between the saturation concentration of a pure species (C*i =C*0,i), number of carbons per molecule, and number of oxygens per molecule (nO) ignoring sulfate and nitrate for use with the 2-D volatility basis set (VBS)…

- Figure 4: I wonder why C* of glyoxal is 1e02 ug m-2. I believe C0 of glyoxal is much higher based on eq(2). Is this because glyoxal is very water soluble so that C* is lowered?

The point labeled "GLY" in Figure 4 corresponds to glyoxal aerosol produced from reaction with OH (Table 2). Since that aerosol is nonvolatile in the model, it is arbitrarily plotted at a C* of 0.01 ug/m3 (as already indicated in figure caption).

- Figure 4: It seems that high molar mass and low volatility compounds (i.e., particlephase products such as dimers, peroxyhemiacetals, oligomers, etc. that would locate close to the alkane line) were not considered in CMAQ. Do you have reason? Were such compounds not detected by measurements in SOAS?

Our model does contain oligomers (OLGB and DIM for example). To be close to the alkane line, that would imply they have little oxygen. The DIM species is a tetrol-tetrol dimer with an OM/OC of 2.07. OLGB (the generic biogenic oligomer) has an OM/OC of 2.1. Thus, the oligomers we have in the model are highly oxygenated and do not fall on the alkane line.  Due to SOAS site being regionally representative and experiencing aged pollution, most compounds should be oxygenated to some degree.

- P13, L12, P14, L2: All IEPOX-derived species were assumed to be non-volatile and I suppose that the model also assumed that such product formation is irreversible. Have you evaluated this assumption by sensitivity simulations?Is it possible that such product formation is actually reversible and might evaporate under certain conditions?

We have not examined the volatility of IEPOX SOA in CMAQ. Recent studies have explored the semivolatile nature of IEPOX SOA through measurements. Lopez-Hilfiker et al. (already cited in manuscript) indicates IEPOX-OA is largely nonvolatile and measured 2-methyltetrols may contain decomposition products of oligomers. Hu et al. (2016) reached the same conclusion using a thermal denuder-AMS combination. While our paper was in review, Isaacman-VanWertz et al. (2016) was also published and indicates that measured 2-methyltetrols, while semivolatile, are more heavily partitioned to the particle phase than their volatility would suggest. We will add a citation to Isaacman-VanWertz et al. We are exploring the reversibility of IEPOX-OA partitioning as part of future work.

- P15, Figure 7: Model reproduced OC well, while overestimating POC and underestimating SOC. Is it possible actually this may suggest that measurements might have overestimated SOC? I suppose that SOC was assumed to be equivalent to OOA based

on AMS-PMF analysis (correct?). What are uncertainties of the measurements? I wonder this, as AMS measures chemical composition and estimate secondary formation processes by post numerical analysis, while the model simulates secondary processes directly.

Two methods were used to estimate the primary/secondary split. (1) For AMS data, PMF analysis provided a BBOA factor at CTR, but no primary factor at LRK. HOA was not resolved at LRK or CTR during SOAS. (2) Section 2.9 presents a modified OC/EC technique for estimating primary OC from routine monitoring networks such as SEARCH, IMPROVE, and CSN. The assumption is that the model POC/EC ratio can be used to estimate observed POC. Since POC is semivolatile, we estimate that a fraction of POC will volatilize in the atmosphere thus reducing the POC/EC ratio downwind of sources. This is uncertain and may be a lower bound on SOC if chemical transformation also converts POC to SOC. However, if emission inventories underestimate SVOC emissions, the POC/EC method may underestimate POC. The two methods are relatively consistent, but there is considerable uncertainty. For that reason, we indicate "Model predictions of OC, SOC, and POC were compared to network observations using the methods described in section 2.9 to determine how model errors in POA (specifically the nonvolatile assumption) could mask errors in SOA."

- P21, L8: Could gas-phase ELVOC formation or particle-phase chemistry help to close the measurement-model gap?

Yes, and this is a topic of future work. Some particle-phase chemistry (acid-catalyzed reactions, oligomerization) is already considered.

References not in the ACPD version:

Gabriel Isaacman-VanWertz, Lindsay D. Yee, Nathan M. Kreisberg, Rebecca Wernis, Joshua A. Moss, Susanne V. Hering, Suzane S. de Sá, Scot T. Martin, M. Lizabeth Alexander, Brett B. Palm, Weiwei Hu, Pedro Campuzano-Jost, Douglas A. Day, Jose L. Jimenez, Matthieu Riva, Jason D. Surratt, Juarez Viegas, Antonio Manzi, Eric Edgerton, Karsten Baumann, Rodrigo Souza, Paulo Artaxo, and Allen H. Goldstein "Ambient Gas-Particle Partitioning of Tracers for Biogenic Oxidation", *Environmental Science & Technology,* 50 (18), 9952-9962, doi: 10.1021/acs.est.6b01674, 2016.

Wagner, N. L., Brock, C. A., Angevine, W. M., Beyersdorf, A., Campuzano-Jost, P., Day, D., de Gouw, J. A., Diskin, G. S., Gordon, T. D., Graus, M. G., Holloway, J. S., Huey, G., Jimenez, J. L., Lack, D. A., Liao, J., Liu, X., Markovic, M. Z., Middlebrook, A. M., Mikoviny, T., Peischl, J., Perring, A. E., Richardson, M. S., Ryerson, T. B., Schwarz, J. P., Warneke, C., Welti, A., Wisthaler, A., Ziemba, L. D., and Murphy, D. M.: In situ vertical profiles of aerosol extinction, mass, and composition over the southeast United States during SENEX and SEAC4RS: observations of a modest aerosol enhancement aloft, Atmos. Chem. Phys., 15, 7085-7102, doi:10.5194/acp-15-7085-2015, 2015.

---

## Referee Comment (RC2) · Anonymous Referee #2 · 16 Nov 2016

Review Pye et al., 2016

This study uses the regional chemical transport model CMAQ to predict the organic aerosol loading and water uptake over the eastern United States and compares the model simulations with observations from monitoring networks. A parameterization of the water uptake by the organic fraction of the aerosol particles is introduced, which should account for the positive feedback of water uptake triggering additional partitioning of semivolatile species to the particulate matter. Simulations relying on different parameterization of water uptake are compared with each other and with observations. It is found that the loading and water uptake of organic aerosol is sensitive to aerosol water interactions with semivolatile organic aerosol species. From too low to too high aerosol mass is predicted depending on how the water uptake is parameterized.

The subject of this study is timely and important. Knowledge of the physical properties of the organic aerosol is increasing, rendering a more physical representation in models feasible. While this paper goes into the right direction, there are inadequacies in the way the water uptake is parameterized. Most importantly, the water uptake in the case of liquid-liquid phase separation (LLPS) is unrealistic because it assumes that in the absence of LLPS the water associated with the inorganic ions is available for partitioning of organic species, while the water uptake of the organics themselves and solution non-ideality are neglected. However, LLPS results from solution non-ideality. The assumptions for the simulation LLPS are therefore inconsistent and insinuate a wrong perception of LLPS, which might lead to confusion. While it makes sense to use the formula by You et al. (2013) to estimate how frequently LLPS in the eastern US occurs (as shown in Fig. 10), the water uptake in the presence of LLPS cannot be calculated the way it is done in the manuscript. This simulation should therefore be removed. Zuend and Seinfeld (2012) have shown that in the case of LLPS, it suffices to assume that the phase separation into an organic and inorganic phase is complete, however, when organics and inorganic constituents are present in the same phase, activity coefficients accounting for interactions between organic and inorganic species should not be neglected as is done in all simulations presented here. Nevertheless, this study goes into the right direction and highlights the need to include solution non-idealities for further model improvements. I therefore recommend this paper for publication in ACP after the following comments have been addressed.

General comment:

It is difficult to find orientation in the manuscript. Some information is scattered, some is incomplete and some seems to be missing. Many different parameters are used. Their meanings are only given once when they are introduced and it is difficult to find this position again. A table listing all abbreviations would therefore be helpful. Also, some parameters and expressions are not used consistently throughout the manuscript (see specific comments). There is quite a bit of supporting information provided, but only partly referenced in the main text, which makes it hard to make use of it. The text in the figure captions is difficult to follow and needs to be improved.

Specific comments:

Page 6, line 12 and 13: the exponent of the equation does not seem to be correct: should it be "63,000(298K-T)/298K"? Please correct.

Page 6, line 18: was the accuracy for Henry's law coefficients determined for dilute solutions or for concentrated solutions as present in aerosol particles? The authors should comment on the accuracy of Henry's law coefficients applied to concentrated solutions using equation (20) and compare it to the accuracy reached by direct use of activity coefficients given by $\gamma$.

Page 8, equation (9): how is $D_{core}$ determined?

Page 9 line 2: what is meant by "mole-based"? Activity coefficients of organics are usually mole fraction based.

Page 9, line 15 and 16: "Inclusion of water, and even inorganic constituents, in the absorbing phase has been encouraged for simplified models in order to reproduce more detailed calculations (Zuend et al., 2010)." Yes, but at the same time also solution non-ideality has to be included, otherwise water uptake is too high. Zuend and Seinfeld (2012) state in their conclusions: "However, assuming ideality at higher RH (>60 %), will very likely lead to significant overprediction of SOA mass and total PM mass." This is the case for organic mixtures and even more for mixtures including inorganic salts. Therefore, this sentence cannot be used as a justification for the model assumptions.

Page 9, line 27: it should be "modified Raoult's law" (as in Zuend et al., 2010, and Zuend and Seinfeld, 2012) because, strictly, "Raoult's law" does not include the activity coefficient $\gamma$.

Page 10, equation (20): the one-constant Margules equation is applicable to molecules of the same molecular size but different polarity. The molecular size of organic species and water is very different. Have you validated the applicability of this equation to aqueous organic solutions?

Page 11, line 25: Figure 3 should be explained better, here or in the figure caption. The meaning of the dashed arrow should be given explicitly.

Page 11, line 27: as explained in the general comment, the assumptions of the simulation LLPS are unphysical and inconsistent. Therefore, this simulation should be removed. It does not represent the water uptake of a system with LLPS.

Page 11, line 30: The assumptions of the second simulation should be explained better by referring to the relevant equations.

Page 12, line 3: The meaning of "a posteriori parameters" should be explained better. How were they determined?

Page 12, line 9 and 10: "The total nonvolatile POA in CMAQ is assumed to correspond to emissions of $C_i^* \approx 3000 \ \mu gm^{-3}$ and lower volatility compounds." I am not sure whether I understand this sentence correctly. Are emissions with $C_i^* \approx 3000 \ \mu gm^{-3}$ considered as nonvolatile? Please explain.

Page 12, line 10 and 11: This sentence should be formulated better.

Page 12, equation (27): what is $\alpha_i$?

Page 12, line 15: volatility bins are defined from $0 - 1000 \ \mu gm^{-3}$. What is meant by the $0 \ \mu gm^{-3}$ volatility bin? Really nonvolatile? Moreover, volatility bins do not directly refer to species. This sentence has to be reformulated.

Page 13 and Fig. 4: How is saturation concentration determined in this figure? Is it the one of the pure compound? In this case, it should be labelled with a zero.

Page 13 and Table 2: The chemical structure is not given in the manuscript. Do the species listed in Table 2 have a specific composition or only physical properties? This should be explained better.

Page 15, line28: the definition of "NMB" and "mean absolute gross error" etc. should be given in the main manuscript or in the SI.

Page 16, line 16: I state here my main criticism again: implementing LLPS just by using the LLPS threshold from You et al. (2013) does not make sense. Water associated to ions is not available for partitioning of organics. The LLPS simulation should therefore be removed.

Page 16, lines 30 – 33: "Generally, all aerosol water is expected to evaporate in an aerodynamic lens inlet used on many instruments (Zelenyuk et al., 2006), which can cause changes in the aerosol phase state (Pajunoja et al., 2016) and could potentially lead to changes in partitioning of soluble organic compounds." Could this be the reason for the absence of the diurnal cycle in the observations? This would be an interesting point to discuss.

Page 17, Figure 10: Figure 10 should be shown also when the LLPS simulations is removed. The prediction of the occurrence of LLPS from You et al. (2013) is based on RH information and shows how relevant LLPS is in the eastern US.

Page 17, lines 4 – 7 and 20 – 23: As stated before, water uptake and semivolatile organic gas-particle partitioning including LLPS can only be modeled realistically when activity coefficients between organics and inorganic ions are included.

Page 20, lines 9 and 10: it should be explicitly stated what precursors are meant.

Page 20, line 27: what do the numbers refer to? To the different simulations or the different observations?

Page 21, lines 14 to 15: Is the second conclusion referring to the simulation $W_o > 1$? This simulation led to the largest overprediction of OA and LWC in Fig. 11! How can this be considered as the best choice?

Figure 1: this figure caption is confusing and needs improvement. The categories are not explained properly. What is "insoluble"? Can you tell how it is derived maybe by referring to an equation? The expression "other SOA" is only used in Fig. 1. Is it the same as "dry organic aerosol" from Table 2? What is "aq. SOA"? The same as "aqueous aerosol" in Table 2 or does it also include cloud water from Table 2? According to Fig. 2, there seem to be two POA species (POC and NCOM). They are considered as nonvolatile. How does this go along with the sentence: "The total nonvolatile POA in CMAQ is assumed to correspond to emissions of $C_i^* \approx 3000\ \mu gm^{-3}$ and lower volatility compounds."? This seems to imply that there is also a semivolatile POA. Is POA considered to be insoluble in water? If yes, this would be inconsistent with the water uptake associated with POA as sketched in Fig. 3.

Figure 2: the meaning of dashed and solid arrows should be stated explicitly.

Figure 3: Are the pie-charts intended to represent the contributions of the different categories to the total particulate matter realistically? Is "$P^{vap}SOA$" in Fig. 3 the same as "dry organic aerosol" in Table 2? What is meant by interaction via acidity? What is hidden within the "etc"?

Figure 4: The quality of the figure needs to be improved. The labels are on top of each other. Some of the species with large OM/OC are listed as the "dry organic aerosol" that does not seem to be involved in the uptake of water. Is this correct? If yes, why?

Technical comments:

Figures are not numbered consecutively as they appear in the text: Figure 4 is mentioned on page 5 while Fig. 3 only on page 11. Figures should therefore be reordered.

Page 4, line 15: remove first "aerosol"

Page 11; line 6: should this sentence read: "…in phase a compared to the total particulate species…"?

Page 12, line 9: "The volatility distribution of gasoline vehicle POA from May et al. (2013) and used by the CMAQ-VBS (Koo et al., 2014) was…" improve formulation.

Page 12, line 15: should it be "0.01, 0.1, 1, 10, 100, …"? Moreover, volatility bins do not directly refer to species. This sentence needs to be reformulated.

Page 14, line 6: "…not entirely…": add "be".

Page 15, line 20: "The standard deviation (s) in model predicted SOA fraction was much higher at 0.21 vs 0.08 in observations." Improve formulation.

Page 16, lines 15 and 16: "Figure 9 shows how including water interactions in absorptive partitioning calculations affected model performance compared to routine monitoring networks." Improve formulation.

Page 17, line 14: "occurred" should be replaced by "is predicted".

Page 17, line 19: the "ed" should be removed from "represented".

Table 1: Is aqueous aerosol used synonymous with aqueous SOA? if yes, only one expression should be used.

Table 2: all abbreviations should be given in the table caption or a new table with all abbreviations should be added.

---

## Author Comment (AC2) · 23 Nov 2016

"On the implications of aerosol liquid water and phase separation for organic aerosol mass"
by Havala O. T. Pye et al.

Response to reviewer #2

We thank reviewer #2 for their comments, editorial corrections, and overall recommendation for publication after comments have been addressed. Our responses are in blue with new text added to the manuscript underlined.

Review Pye et al., 2016
This study uses the regional chemical transport model CMAQ to predict the organic aerosol loading and water uptake over the eastern United States and compares the model simulations with observations from monitoring networks. A parameterization of the water uptake by the organic fraction of the aerosol particles is introduced, which should account for the positive feedback of water uptake triggering additional partitioning of semivolatile species to the particulate matter. Simulations relying on different parameterization of water uptake are compared with each other and with observations. It is found that the loading and water uptake of organic aerosol is sensitive to aerosol water interactions with semivolatile organic aerosol species. From too low to too high aerosol mass is predicted depending on how the water uptake is parameterized.

The subject of this study is timely and important. Knowledge of the physical properties of the organic aerosol is increasing, rendering a more physical representation in models feasible. While this paper goes into the right direction, there are inadequacies in the way the water uptake is parameterized. Most importantly, the water uptake in the case of liquid-liquid phase separation (LLPS) is unrealistic because it assumes that in the absence of LLPS the water associated with the inorganic ions is available for partitioning of organic species, while the water uptake of the organics themselves and solution non-ideality are neglected. However, LLPS results from solution non-ideality. The assumptions for the simulation LLPS are therefore inconsistent and insinuate a wrong perception of LLPS, which might lead to confusion. While it makes sense to use the formula by You et al. (2013) to estimate how frequently LLPS in the eastern US occurs (as shown in Fig. 10), the water uptake in the presence of LLPS cannot be calculated the way it is done in the manuscript. This simulation should therefore be removed. Zuend and Seinfeld (2012) have shown that in the case of LLPS, it suffices to assume that the phase separation into an organic and inorganic phase is complete, however, when organics and inorganic constituents are present in the same phase, activity coefficients accounting for interactions between organic and inorganic species should not be neglected as is done in all simulations presented here. Nevertheless, this study goes into the right direction and highlights the need to include solution non-idealities for further model improvements. I therefore recommend this paper for publication in ACP after the following comments have been addressed.

For the LLPS simulation, water uptake is calculated using ISORROPIA II, which is also used to calculate aerosol water in standard CMAQ. Given that inorganic and organic constituents likely mix in the ambient atmosphere (Figure 10b), even the base model is inaccurate in its representation of aerosol water as it neglects water uptake onto organic constituents and inorganic-organic interactions that would affect water associated with inorganic constituents. LLPS and the base simulation underestimate aerosol liquid water (Figure 11e).

One can argue that, although LLPS is caused by nonidealities (i.e., high activity coefficients), once the phases have separated (or come together), the constituents in each are in a relatively favorable state and could behave in their respective phase in an "ideal" way - and in equilibrium across phases. By using the work of You et al. 2013 for predicting when LLPS occurs, we start with a phase state that is relatively favorable before calculating partitioning.

The LLPS simulation is a hybrid of empirical (to represent the phase separation based on OM/OC and RH) and theoretical (specifically ideal Raoult's law) representations of the influence of inorganic water on SOA. To emphasize that the LLPS simulation is meant to separate the effect of "inorganic water" from "organic water" we have renamed it "ideal Wi." Ideal Wi is meant to demonstrate the effects of including water in the partitioning medium for organics without accounting for deviations in ideality which is a useful reference case upon which to build in the future. Some rephrasing is implemented throughout the manuscript to emphasize what is captured by "ideal Wi" (see track changes).

General comment:
It is difficult to find orientation in the manuscript. Some information is scattered, some is incomplete and some seems to be missing. Many different parameters are used. Their meanings are only given once when they are introduced and it is difficult to find this position again. A table listing all abbreviations would therefore be helpful. Also, some parameters and expressions are not used consistently throughout the manuscript (see specific comments). There is quite a bit of supporting information provided, but only partly referenced in the main text, which makes it hard to make use of it. The text in the figure captions is difficult to follow and needs to be improved.

To provide more guidance to the reader, we have revised the last paragraph of the introduction to refer to all second level headings. In addition, the section outlining simulations to be performed has been moved earlier in the manuscript (to section 2.1) to give an overview and outline of the article. Although section 2.7 is listed before 2.6 in the introduction, we kept the order as-is in the manuscript since we wanted to introduced the modified Raoult's law before discussing deviations in ideality. See track changes.

Supporting information was meant to provide additional information for those with interest in specific topics such as model evaluation and description. We do not want the reader to consider the supporting information as mandatory reading but as optional reading, so references to it are kept at a minimum.

See responses to specific comments below. The caption to table 2 now provides one location where many parameters are defined.

Specific comments:
Page 6, line 12 and 13: the exponent of the equation does not seem to be correct: should it be "63,000(298K-T)/298K"? Please correct.

The equation is correct. $H = 4.1 \times 10^3 \exp(63,000(298-T)/(298T))$ is the Van't Hoff equation (see Raventos-Duran et al. 2010 equation 2: http://www.atmos-chem-phys.net/10/7643/2010/acp-10-7643-2010.pdf). The coefficient 63,000 corresponds to the enthalpy of solvation (J/mol) divided by the universal gas constant (J/molK) and thus has units of K. We added the K units to the equation:

$$H = 4.1 \times 10^3 \exp(63,000\underline{K}(298-T)/(298T))$$

Page 6, line 18: was the accuracy for Henry's law coefficients determined for dilute solutions or for concentrated solutions as present in aerosol particles? The authors should comment on the accuracy of Henry's law coefficients applied to concentrated solutions using equation (20) and compare it to the accuracy reached by direct use of activity coefficients given by ⬚.

GROMHE Henry's law coefficients are for dilute solutions, and thus evaluation of the Henry's law coefficient by comparison with other models (HenryWin, etc) neglects the influence of inorganic salts which may increase (salting in) or decrease (salting out) the Henry's law coefficient compared to pure water. The following modifications were made:

> GROMHE was found to reproduce Henry's Law coefficients for organic-water systems with mean absolute error of about 0.3 log units compared to 0.5 for HenryWin and 0.4 for SPARCv4.2 (Raventos-Duran et al., 2010).

> Although the relationship between H and $C^*_i$ was relatively robust, variability in H spanned many orders of magnitude for a given $C^*_i$ bin without considering how inorganic species may modify the Henry's law coefficient.

To get an idea of how Henry's law may be modified in the presence of organics, we refer readers to Figure S2-S3 which show activity coefficients and $C^*$ as a function of mole fraction water.

Page 8, equation (9): how is Dcore determined?

An equation for Dcore is now available right after equation 8. Text after equation 8 indicates Dcore is the volume equivalent accumulation mode diameter excluding water associated with organic species. Notation in old equation 10 (now 11) was updated for consistency. New/updated equations:

$$D_{core} = \left( \frac{6}{\pi} \sum_{i \neq W_o} V_i \right)^{1/3}$$

$$\kappa = \frac{\sum_{i \neq W_o} (\kappa_i V_i)}{\sum_{i \neq W_o} (V_i)}$$

Page 9 line 2: what is meant by "mole-based"? Activity coefficients of organics are usually mole fraction based.

Not all activity coefficients currently used in atmospheric science are mole based. Activity coefficients introduced for the volatility basis set (VBS) are molality based (See Supplemental Material section 5.1 of Donahue et al. 2006 ES&T: http://pubs.acs.org/doi/full/10.1021/es052297c). In addition, ISORROPIA-II uses molal-based activity coefficients (Kusik-Meisner for binary solutions and Bromley's method for multicomponent). We are emphasizing that activity coefficients in this work are mole based.

Page 9, line 15 and 16: "Inclusion of water, and even inorganic constituents, in the absorbing phase has been encouraged for simplified models in order to reproduce more detailed calculations (Zuend et al., 2010)." Yes, but at the same time also solution non-ideality has to be included, otherwise

water uptake is too high. Zuend and Seinfeld (2012) state in their conclusions: "However, assuming ideality at higher RH (>60 %), will very likely lead to significant overprediction of SOA mass and total PM mass." This is the case for organic mixtures and even more for mixtures including inorganic salts. Therefore, this sentence cannot be used as a justification for the model assumptions.

We understand the point raised by the reviewer. However, our approach for water uptake, the ZSR correlation used for the inorganic phase for inorganic models (eg ISORROPIA) and "hygroscopicity" for organic systems does not require explicit calculations of water activity. The latter is required of course to PREDICT LLPS, but we do not do that here. Instead, we parameterize LLPS and predict the water uptake for each phase using, effectively, ZSR in each phase separately, which of course is thermodynamically consistent as water activity is consistently equal to RH in all the phases.

See our earlier response regarding the assumptions of the LLPS (now "ideal Wi") simulation. The fact that organics and water are present in one phase is an indication that conditions cannot be that unfavorable. Our "ideal Wi" simulation is consistent with the conclusion of Zuend and Seinfeld that assuming ideality at higher RH will lead to overpredictions in SOA and we think it is useful to demonstrate this fact in the context of a regional model compared with field data. Also note that the amount of water in the particle predicted in "ideal Wi" is lower than that observed (Figure 10e) and thus water uptake is not too high.

Page 9, line 27: it should be "modified Raoult's law" (as in Zuend et al., 2010, and Zuend and Seinfeld, 2012) because, strictly, "Raoult's law" does not include the activity coefficient $\gamma$.

We agree and have added the term "modified" before Raoult's at two locations in the manuscript and added a citation to Seinfeld and Pandis (2006).

Page 10, equation (20): the one-constant Margules equation is applicable to molecules of the same molecular size but different polarity. The molecular size of organic species and water is very different. Have you validated the applicability of this equation to aqueous organic solutions?

The one-constant Margules is a simple model and we have chosen it for this reason. As part of future work, we have begun exploring if ambient partitioning of individual compounds can be described using a Margules model and preliminary results indicate it performs better in terms of capturing variability in particle fraction than ideal partitioning to an organic-only mixture for some species. In the near-future, the constant in the Margules model could potentially be determined empirically via regression as has been done for the volatility of SOA in the Odum 2-product or VBS parameterizations. In addition, we are exploring other models for activity coefficients.

Page 11, line 25: Figure 3 should be explained better, here or in the figure caption. The meaning of the dashed arrow should be given explicitly.

The caption indicates: "The white dashed arrows indicate aqueous SOA interaction with the inorganic phase (via liquid water, acidity, etc)." etc has been replaced with "particle size" in response to a later comment. The figure is illustrative (see later comments about size of pie slices).

Page 11, line 27: as explained in the general comment, the assumptions of the simulation LLPS are unphysical and inconsistent. Therefore, this simulation should be removed. It does not represent the

water uptake of a system with LLPS.

See earlier response regarding LLPS/"ideal Wi."

Page 11, line 30: The assumptions of the second simulation should be explained better by referring to the relevant equations.

We now refer back to section 2.5 which contains all the equations used to predict Wo.

> In the second simulation, uptake of water to the organic phase (Wo > 0) was predicted based on its OM/OC and k-Köhler theory (Petters and Kreidenweis, 2007) (Section 2.5).

Page 12, line 3: The meaning of "a posteriori parameters" should be explained better. How were they determined?

A posteriori parameters are introduced in section 2.6 where we state that a priori assumptions resulted in all particulate organic nitrates being driven out of the particle. We performed the following series of simulations until organic nitrates returned to the particle:

| Sensitivity | MTNO3 H-law value [M/atm] | Change in Activity Coefficients | Adequate organic nitrates in particle at SOAS-CTR? |
|---|---|---|---|
| 0601 | 1.5e6 | Base | No |
| 0610 | 1.5e7 | Base | No |
| 0613 | 1.5e8 | Base | No |
| 0616 | 1.5e8 | Base/10 | Yes |

Text has been added:

> These adjustments, determined through a series of sensitivity simulations, may have been necessary due to inaccuracies in the Henry's law coefficients, pure species saturation concentrations, molecular weights, Margules model, or a combination of all of the above.

> A posteriori parameters used in $\gamma \neq 1$, which include a factor of 100 increase in MTNO3 solubility and factor of 10 decrease in activity coefficients, are available in Table S6.

Page 12, line 9 and 10: "The total nonvolatile POA in CMAQ is assumed to correspond to emissions of $C_i^* \approx 3000$ μgm−3 and lower volatility compounds." I am not sure whether I understand this sentence correctly. Are emissions with $C_i^* \approx 3000$ μgm−3 considered as nonvolatile? Please explain.

The National Emissions Inventory does not estimate the volatility of primary organic aerosol (POA), however CMAQ assumes it is nonvolatile. We assume that the model POA corresponds to species with $C_i^* \approx 3000$ μg m$^{-3}$ and lower volatility compounds. The assumption that POA corresponds to semivolatile species is only made in post-processing for model evaluation. Revision:

> For estimating observed POA from total OA only,  POA in CMAQ is assumed to correspond to emissions of $C_i^* \approx 3000$ μg m$^{-3}$ and lower volatility compounds.

Page 12, line 10 and 11: This sentence should be formulated better.

See previous word removal.

Page 12, equation (27): what is $\alpha i$?
Addressed with next comment.

Page 12, line 15: volatility bins are defined from 0 – 1000 µgm-3. What is meant by the 0 µgm-3 volatility bin? Really nonvolatile? Moreover, volatility bins do not directly refer to species. This sentence has to be reformulated.
0 indicated nonvolatile for all atmospheric conditions (entirely in particle). The sentence has been reworded.

> …where the volatility profile is described by one nonvolatile and $C_i^*$ =  1, 10, 100, and 1000 surrogate species in the following mass-based abundance ($\alpha_i$): 0.27, 0.15, 0.26, 0.16, and 0.17.

Page 13 and Fig. 4: How is saturation concentration determined in this figure? Is it the one of the pure compound? In this case, it should be labelled with a zero.

It is the pure species saturation concentration (usually from an Odum 2-product fit). Text was updated and the x-axis label updated to include a subscript 0 for pure species:

> Figure 4 shows the updated molecular weights as a function of pure species saturation concentration and colored by OM/OC.

Page 13 and Table 2: The chemical structure is not given in the manuscript. Do the species listed in Table 2 have a specific composition or only physical properties? This should be explained better.

Some species have a specific composition while others have reasonable physical properties considering their parent hydrocarbon. Table 1 gives a species name where a specific species is represented. The heading of column 2 in Table 1 was modified from "Production pathway description" to "Species or production pathway description".

Page 15, line28: the definition of "NMB" and "mean absolute gross error" etc. should be given in the main manuscript or in the SI.

MB, ME, and NMB formulas are now given in the caption of Figure 7.

**Figure 7.** Aerosol OC, POC, and SOC predicted by the base model simulation ($M_i$) compared to CSN, IMPROVE, and SEARCH (JST, BHM, CTR, and YRK) observations ($O_i$). Mean bias ($MB = \frac{1}{n}\Sigma_{i=1}^{n}(M_i - O_i)$) and mean absolute gross error ($ME = \frac{1}{n}\Sigma_{i=1}^{n}|M_i - O_i|$) are in µgCm$^{-3}$. X symbols indicate mean bias. Boxplots indicate 5th, 25th, median, 75th, and 95th percentile. $r^2$ based on a zero intercept. $n$ is the number of observations. $NMB = \frac{\Sigma_{i=1}^{n}(M_i - O_i)}{\Sigma_{i=1}^{n} O_i}$.

Page 16, line 16: I state here my main criticism again: implementing LLPS just by using the LLPS threshold from You et al. (2013) does not make sense. Water associated to ions is not available for partitioning of organics. The LLPS simulation should therefore be removed.

See earlier renaming of LLPS to "ideal Wi." The water associated with ions would be available for partitioning if one homogeneous phase existed in the particle (ie SRH<RH).

Page 16, lines 30 – 33: "Generally, all aerosol water is expected to evaporate in an aerodynamic lens inlet used on many instruments (Zelenyuk et al., 2006), which can cause changes in the aerosol phase state (Pajunoja et al., 2016) and could potentially lead to changes in partitioning of soluble organic compounds." Could this be the reason for the absence of the diurnal cycle in the observations? This would be an interesting point to discuss.

Co-located hourly SEARCH measurements of OC (Figure S1), which do not use aerodynamic lenses, show a similar lack of strong diurnal variation as the AMS. Thus it is unlikely that the diurnal variation was affected by the AMS lens. Biases of course are always possible and e.g., SEARCH measurements use a thermal-optical method to separate OC from EC and may be subject to their own artifacts.

While water can be lost in the aerodynamic lens, any organic species that are partitioned to the aerosol due to volatility will have vapor pressures at least 5 orders of magnitude lower. Evaporation rate is proportional to vapor pressure, and since water evaporates at a rate of $\sim 10^4$ monolayers s$^{-1}$, semivolatile species can be estimated to evaporate at a rate of 0.1 monolayers s$^{-1}$, or 0.001 monolayers in the 10 ms residence time in the AMS aerodynamic lens. This is consistent with the widely-observed lack of evaporation of ammonium nitrate, a semivolatile species, in the AMS lens (Canagaratna et al., 2007). If the aerosol would transition to a glassy state during rapid water evaporation, vapor pressure of organics would overall be reduced by orders of magnitude, which tends to mitigate volatilization biases.

If the ambient is much colder than the instrument operation temperature, the heating can increase vapor pressure enough (even if the aerosol is glassy) and perhaps promote volatilization biases. This situation was infrequent during SOAS, when most instruments were located in air-conditioned trailers (~20-25°C) at temperatures similar or often lower than ambient. For the infrequent conditions in which trailers were warmer than ambient air, we can use information from the literature to provide a constraint on this possible effect. Guo et al. (2016) estimated whether volatilization biases associated with measurements ammonium nitrate (a soluble and highly semivolatile species) were present under aircraft operation during the WINTER campaign (when the air sampled was 15-40°C colder than the instrumentation in the cabin of the aircraft). NO$_3$ concentrations, as measured by a PILS and an AMS as well as thermodynamic partitioning calculations indicate volatilization biases were not present - in the AMS aerodynamic lens as well. Many organic species have low volatility (e.g. Cappa and Jimenez, 2010). For organic species that are comparable to NO$_3$ (semivolatile, soluble in water, and with high molecular diffusivity) we would not expect issues in the measurements for the reasons described above.

A related issue, however, is that organic compounds with very high vapor pressures (approaching that of water) and water solubility. We have added/revised the following text to address this point:

> Some caution should be applied when comparing model predictions and observations. Measurements of total aerosol mass from IMPROVE and CSN networks are made under relative humidities of 30-50\% and quartz filters for OC analysis from IMPROVE may be subject to ambient conditions in the field and during shipping before analysis (Solomon et al. 2014). Exposure to low RH could cause evaporation of reversible aqueous SOA (El-Sayed et al. 2016). Kim et al. (2015) have  reported the IMPROVE measurements of OC were 27% lower than colocated SEARCH measurements during the summer of 2013, and hypothesized the difference to be due to evaporation from the IMPROVE filters, during and after sampling artifacts. Episodic field campaign observations may be subject to sampling biases as well. Dryers are used ahead of many online aerosol chemistry instruments, and  most

aerosol water is expected to evaporate in an aerodynamic lens inlet used on many instruments (Zelenyuk et al. 2006; Matthew et al., 2009). Such drying  can cause changes in the aerosol phase state (Pajunoja e tal. 2016) and could potentially lead to changes in partitioning of soluble organic compounds. El-Sayed et al. (2016) have reported a loss of WSOC after trying. Those authors used a post-drying residence time of 7 s, which is much longer than those used post-drying for the AMS in SOAS (~1 s) and the time in the aerodynamic lens (~0.01 s). A prior study reported that evaporation of ammonium nitrate, a water soluble and semivolatile species, was not observed when using post-drying residence times of ~1 s (Guo et al., 2016). While this topic should be subject to additional research, the AMS data in SOAS is unlikely to have significant biases due to this effect.

Page 17, Figure 10: Figure 10 should be shown also when the LLPS simulations is removed. The prediction of the occurrence of LLPS from You et al. (2013) is based on RH information and shows how relevant LLPS is in the eastern US.

We have kept figure 10 as it shows useful information on the frequency of time particles are present as one-phase vs 2-phases. The prediction of that occurrence is empirical as estimated by You et al. and is not strongly influenced by the different sensitivity simulations (their only influence would be through changes in the OM/OC which are not very large, Figure 11c). The caption has been reworded to go along with the new characterization of the LLPS simulation as "ideal Wi." The LLPS/ideal Wi simulations did consider information in Figure 10.

Page 17, lines 4 – 7 and 20 – 23: As stated before, water uptake and semivolatile organic gas-particle partitioning including LLPS can only be modeled realistically when activity coefficients between organics and inorganic ions are included.

See earlier responses regarding "ideal Wi." We have removed the text indicating LLPS is a lower bound on the effect of inorganic water on semivolatile OA.

>

Page 20, lines 9 and 10: it should be explicitly stated what precursors are meant.

Revised:

> Figure 14 shows observed water soluble organic carbon compounds in the gas phase (WSOCg, measured by Mist Chamber and Total Carbon Analzyer Hennigan et al. 2009) compared to (a) semivolatile SOA precursors (i.e. those associated with dry organic aerosol in Table 1) and (b) semivolatile and aqueous SOA precursors currently in CMAQ.

Page 20, line 27: what do the numbers refer to? To the different simulations or the different observations?

Numbers refer to different observations (networks). The single simulation name is now included for clarification:

A method (γ≠1 simulation) was developed to take into account deviations from ideality using an activity coefficient calculated based on the species Henry's law coefficient, pure species saturation concentration … and mole fraction water in the particle that resulted in a normalized mean bias of -4%, -10%, and -2% for IMPROVE, CSN, and SEARCH SOC.

Page 21, lines 14 to 15: Is the second conclusion referring to the simulation Wo > 1? This simulation led to the largest overprediction of OA and LWC in Fig. 11! How can this be considered as the best choice?

It refers to the fact that OM/OC is providing some indication of kappas (Figure 11c-d). While there is still progress to be made, neglecting water associated with organics (see Figure 11e) is wrong with implications for any process influenced by particle size (deposition, light scattering, etc).

Figure 1: this figure caption is confusing and needs improvement. The categories are not explained properly. What is "insoluble"? Can you tell how it is derived maybe by referring to an equation? The expression "other SOA" is only used in Fig. 1. Is it the same as "dry organic aerosol" from Table 2? What is "aq. SOA"? The same as "aqueous aerosol" in Table 2 or does it also include cloud water from Table 2? According to Fig. 2, there seem to be two POA species (POC and NCOM). They are considered as nonvolatile. How does this go along with the sentence: "The total nonvolatile POA in CMAQ is assumed to correspond to emissions of $C_i^* \approx 3000$ µgm−3 and lower volatility compounds."? This seems to imply that there is also a semivolatile POA. Is POA considered to be insoluble in water? If yes, this would be inconsistent with the water uptake associated with POA as sketched in Fig. 3.

Insoluble is total OA minus 2.1xWSOC. Other SOA is the same as dry organic aerosol in table 2. Aq. SOA is the same as aqueous SOA and was abbreviated to keep the figure sized properly. POA does not take up water in the base simulation (standard CMAQ), but does in the Wo>0 and nonideal simulation (Figure 3 is correct). See also earlier revision regarding semivolatile POA.

Modifications:

Caption: Contribution of POA (observed biomass burning OA, BBOA, Xu et al. (2015a)), SOA, water-soluble OA (estimated as 2.1 x WSOC from the PiLS, Sullivan et al. (2004)), and aqueous (aq.) SOA (model only) to total OA during June 2013 observed at CTR during SOAS and modeled by standard CMAQ. Insoluble OA is the difference between measured total OA and water-soluble OA. Modeled "other SOA" is formed via partitioning to a dry organic phase.

Added to Figure 2 caption:

POA = POC + NCOM.

Figure 2: the meaning of dashed and solid arrows should be stated explicitly.

All arrows are now solid.

Figure 3: Are the pie-charts intended to represent the contributions of the different categories to the total particulate matter realistically? Is "PvapSOA" in Fig. 3 the same as "dry organic aerosol" in Table 2? What is meant by interaction via acidity? What is hidden within the "etc"?

The relative sizes of the pie slices were determined by using SOAS measured concentrations of sulfate-nitrate-ammonium (inorganic phase), BBOA (POA), WSOM (aqueous SOA), and total SOA minus WSOM (Pvap SOA). As we demonstrate in Figure 13 and in the manuscript, measured WSOC (or WSOM) is not strictly a proxy for aqueous SOA. The size of the Pvap vs. aqueous SOA slices may not be correct, and for this reason, we have used the pie figures as illustrative without specifically discussing the magnitudes. The model distribution of POA vs SOA is different than the ambient data as illustrated in Figure 1, thus we didn't want to create pie charts based on model results. There is no atmospheric measure of "dry organic aerosol."

Interaction via acidity indicates the SOA formed from IEPOX which requires acidity. Acidity is estimated via ISORROPIA and thus only considers inorganic contributions to H+. The main thing embedded in the "etc" are the ways inorganic species physically interact to determine IEPOX SOA:

- inorganic species influence the volume of accumulation mode aerosol which is used to convert the concentration of acids and nucleophiles in mol/volume-air to mol/volume-particle which is what governs the rate of particle-phase reaction
- inorganic species affect the surface area of accumulation mode aerosol which is used to calculate heterogeneous uptake

These influences were too verbose to discuss in a figure caption; however, we have replaced "etc" with particle size.

Figure 4: The quality of the figure needs to be improved. The labels are on top of each other. Some of the species with large OM/OC are listed as the "dry organic aerosol" that does not seem to be involved in the uptake of water. Is this correct? If yes, why?

"dry organic aerosol" refers to base model assumptions (no water uptake). In Wo>0 and the nonideal simulation, all organic aerosol takes up water according to its OM/OC. Labels are now separated.

[Figure]

Technical comments:
Figures are not numbered consecutively as they appear in the text: Figure 4 is mentioned on page 5 while Fig. 3 only on page 11. Figures should therefore be reordered.
Figures 1-4 are now in order.

Page 4, line 15: remove first "aerosol"

Removed.

Page 11; line 6: should this sentence read: "…in phase a compared to the total particulate species…"?
Yes, revised.

Page 12, line 9: "The volatility distribution of gasoline vehicle POA from May et al. (2013) and used by the CMAQ-VBS (Koo et al., 2014) was…" improve formulation.
Reworded in response to previous comment.

Page 12, line 15: should it be "0.01, 0.1, 1, 10, 100, …"? Moreover, volatility bins do not directly refer to species. This sentence needs to be reformulated.
Reworded in response to previous comment.

Page 14, line 6: "…not entirely…": add "be".
Added.

Page 15, line 20: "The standard deviation (s) in model predicted SOA fraction was much higher at 0.21 vs 0.08 in observations." Improve formulation.
Sentence replaced with:
    The variability in predicted SOA fraction (standard deviation, s, of 0.21) was much higher than the variability in observed SOA fraction (s=0.08).

Page 16, lines 15 and 16: "Figure 9 shows how including water interactions in absorptive partitioning calculations affected model performance compared to routine monitoring networks." Improve formulation.
    Figure 9 shows how including water interactions in absorptive partitioning calculations affected model predictions of OC at routine monitoring network locations.

Page 17, line 14: "occurred" should be replaced by "is predicted".
Replaced with was predicted.

Page 17, line 19: the "ed" should be removed from "represented".
Removed

Table 1: Is aqueous aerosol used synonymous with aqueous SOA? if yes, only one expression should be used.
Aqueous SOA terminology is not used in Table 1. Aqueous SOA would be a subset of aqueous aerosol which would also include inorganic species.

Table 2: all abbreviations should be given in the table caption or a new table with all abbreviations should be added.
All column headings in Table 2 are now defined in the figure caption.

Other updates to manuscript:
  • CMAQv5.2-beta is currently available on github. CMAQv5.2 will be fully released in 2017. The data availability section has been updated.

- Appel et al. is now in GMDD (citation updated)
- Feiner et al. 2016 is now published (citation updated)

**New references:**

[revised manuscript text omitted]